# REPRESENTATION GAP: EXPLAINING THE UNREASONABLE EFFECTIVENESS OF NEURAL NETWORKS FROM A GEOMETRIC PERSPECTIVE

## ABSTRACT

Understanding generalization is a central issue in machine learning. Recent work has identified two key mechanisms to explain it: the strong memorization capabilities of neural networks, and the task-aligned invariants imposed by their architecture and training procedure. Remarkably, it is possible to characterize the output of a neural network for some classes of invariants widely used in practice. Leveraging this characterization, we introduce the representation gap, a metric that generalizes empirical risk and enables asymptotic analysis across three common settings: (i) unconditional generative modeling, where we obtain a precise asymptotic equivalent; (ii) supervised prediction; and (iii) ambiguous prediction tasks. A central outcome is that generalization is governed by a single parameter – the intrinsic dimension of the task – which captures task difficulty. As a corollary, we prove that popular strategies such as equivariant architectures improve performance by explicitly reducing this intrinsic dimension.

## 1 INTRODUCTION

Implicit specification through data gives neural networks a flexibility that has been leveraged by recent advances to achieve beyond-human performance on a wide spectrum of tasks (Jumper et al., 2021; Ramesh et al., 2021; Silver et al., 2016). Considering unlimited access to data, such neural networks could theoretically learn to solve any data-driven task (Hornik, 1991; Kaplan et al., 2020b). However, apart from some specific cases (e.g., simulated environments), data is costly to gather and process (Deng et al., 2009; Su et al., 2012) and available only in finite amounts. In order to make the most out of available data, practitioners have proposed many techniques to introduce external knowledge in neural network training. This includes neural network architecture with structural invariants (Krizhevsky et al., 2017; Cohen & Welling, 2016), optimization algorithms with task-aligned biases, latent space reparameterization (Engel et al., 2020), or explicit regularization losses (Hoerl & Kennard, 1970; Tibshirani, 1996). A central question in machine learning is to understand how these design choices affect the behavior of a neural network outside the training dataset. While the full understanding of neural network generalization is still an open question, a recent work has identified two key mechanisms to explain it. Firstly, their flexibility to fit arbitrary datasets, and secondly the invariants that are enforced by their design choices (Hornik, 1991; Zhang et al., 2021).

On one hand, recent work on the implicit regularization of gradient descent has suggested that neural networks act as minimal-norm interpolators of the training data (Zhang et al., 2021; Li & Wei, 2021). For instance, linear and kernel regression have been shown to converge to minimal $\mathcal{L}_2$ norm interpolators (Liang & Rakhlin, 2018; Mei & Montanari, 2022; Hastie et al., 2022), while boosting and matrix-factorization algorithm are examples for the $\mathcal{L}_1$ norm (Liang & Sur, 2022; Gunasekar et al., 2018), and stochastic gradient descent favorizes the Sobolev seminorms (Ma & Ying, 2021). This property has been used to explain the strong generalization capabilities of these algorithms (Zhang et al., 2021), the surprising effectiveness of over-parametrization (Allen-Zhu et al., 2019; Belkin, 2021), or the recently observed double-descent phenomenon (Belkin et al., 2019).

On the other hand, recent work on diffusion models has identified the key role played by network architectures and their structural constraints to explain their impressive creativity. Remarkably, the

authors of (Kamb & Ganguli, 2025) have even proposed a closed-form expression predicting with high accuracy the output of a trained model in the setting of convolutional diffusion models.

Crucially, it is possible in both cases to characterize the output of a trained model. In particular, we can completely describe the set $\Omega_f$ of points $(x, f(x))$ that are reachable by a model $f$. Based on this observation, we depart from the usual definition of generalization based on VC-dimension (Vapnik & Chervonenkis, 1971) and Rademacher complexity (Bartlett & Mendelson, 2001) and argue for a geometric perspective instead, as has also been suggested by recent empirical evidence (Stephenson et al., 2021). More precisely, we study the discrepancy between the manifold $\Omega$ from which training data is drawn, and its representation $\Omega_f$ learned by a model $f$. This quantity, that we name *representation gap*, is the focus of our present work. Critically, our analysis do not rely on any assumption about the data distribution $\rho$, but only on the geometry of the manifold $\Omega$ on which this distribution is supported.

We focus on the asymptotic evolution of the representation gap $\mathcal{R}_n$ when the size $n$ of the training dataset $\mathbb{D}$ grows to infinity. We show that this representation gap has a surprisingly simple asymptotic evolution in $n^{-2/d}$, where $d$ is an *intrinsic dimension* parameter that only depends on the geometry of the data manifold $\Omega$ and the symmetries of the model $f$. Remarkably, we show as a corollary that popular techniques used by practitioners to improve model performance, such as the use of equivariance architecture, are in fact reducing this intrinsic dimension $d$ – thereby provably improving performance. This provides a precise and systematic tool to characterize the impact of architecture choice and training procedure design on model performance, data consumption, and task simplification. We validate the predictions of our theory with extensive evaluation over synthetic data as well as real-world data.

In the present work, we make the following contributions.

**We introduce the representation gap**, a generalization of the empirical risk, and analyze its asymptotic behavior across three common settings: (i) unconditional generative modeling, where we obtain a precise asymptotic equivalent; (ii) supervised prediction; and (iii) ambiguous prediction task.

**We show that generalization is governed by the intrinsic dimension of the task**, a single parameter which captures the difficulty level of the task, and may be directly linked to the data manifold geometry and the model invariants. In particular, we show how standard techniques to improve model performance provably reduces this intrinsic dimension.

**We provide experiments** a set of synthetic datasets, which offer controlled test cases for assessing our theoretical results, as well as on the popular MNIST dataset (Lecun et al., 1998).

## 2 RELATED WORK

**Implicit bias of neural network.** Classical analyses of neural networks relied on controlling model complexity to derive generalization bounds (Vapnik & Chervonenkis, 1971; Bartlett & Mendelson, 2001), but such approaches failed to explain the empirical success of over-parametrized deep neural networks. More recent work shows that standard training algorithms tend to converge towards models with low complexity, thereby explaining their strong generalization capabilities (Belkin, 2021; Zhang et al., 2021; Li & Wei, 2021; Allen-Zhu et al., 2019; Belkin et al., 2019). Our analysis is based on this line of work, but we do not make any assumption about the data distribution $\rho$ and adopt a geometric perspective instead. This geometric point of view frees us from positing a fixed data distribution on the manifold – an abstraction that often fail to reflect the nature of real-world data, whether the distribution evolves over time (Kuznetsov & Mohri, 2017), samples are not i.i.d. (Mohri & Rostamizadeh, 2008), or sampling depends on the observer (Settles, 2009). By contrast, assuming that real-world data lie on a manifold is a mild and standard hypothesis, reflecting the structure of many physical systems.

**Geometric perspective on generalization**. Building on the manifold hypothesis (Bengio et al., 2013), several works have shown that neural networks are manifold learners (Loaiza-Ganem et al., 2024; Schuster & Krogh, 2021), while several others have studied the hidden layers topology (Stephenson et al., 2021). Focusing on ReLU networks, the authors of Yao et al. (2024) have derived generalization bounds based on the data manifold characteristics – such as its dimension or Betti number. Our

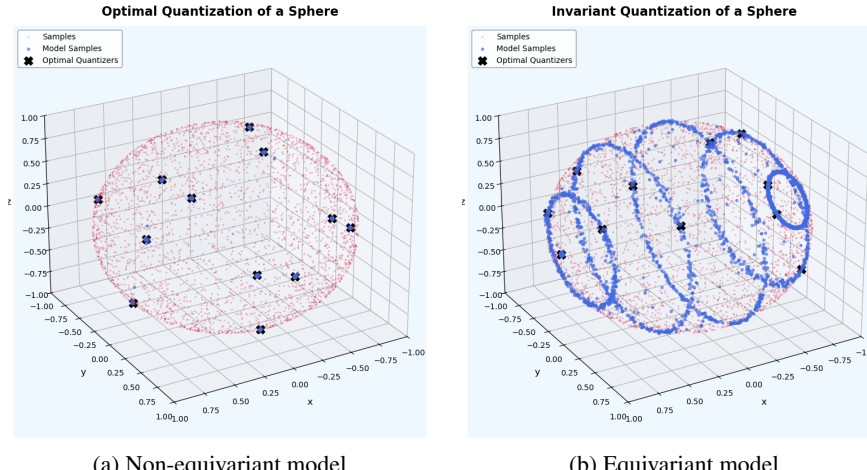

(a) Non-equivariant model         (b) Equivariant model

Figure 1: Illustration of the virtual augmentation of a dataset by an equivariant diffusion model, as well as the corresponding representation gap improvement. Plot (a) shows samples from trained diffusion model, and plot (b) shows samples from a trained equivariant diffusion model (with rotational invariance along axis $x$). In both plots, the shape $\Omega$ is indicated by a dense cloud of red dots, the coarse dataset $\mathbb{D}$ by crosses, and the approximated shape $\Omega_f$ by a dense cloud of blue dots sampled from the trained diffusion model $f$.

work departs from these approaches by providing precise asymptotic equivalents of the models' generalization capabilities.

**Equivariant neural network**. Empirical studies have shown that equivariance improve generalization or sample efficiency (Cohen & Welling, 2016; Bulusu et al., 2022). Closest to our work, the authors of Sannai et al. (2021) established PAC generalization bounds for equivariant and invariant neural networks. In contrast, our analysis provides asymptotic equivalents. Finally, Kamb & Ganguli (2025) derived a closed-form expression for the predictions of trained diffusion models; while our work builds on theirs, we focus on generalization properties rather than generative diversity.

**Scaling Laws**. Our work is closely connected to the Neural Scaling Law literature (Kaplan et al., 2020a), and in particular to recent studies on scaling laws for diffusion models (Mei et al., 2024; Li et al., 2024a; Liang et al., 2024b). However, prior work in Scaling Laws for Diffusion models has primarily focused on scaling with respect to compute, rather than dataset size, which is the focus of our study. Moreover, existing efforts are largely empirical, whereas we provide provable results.

## 3 AN ILLUSTRATIVE EXAMPLE

Let us first introduce the main concepts of this paper with a concrete example. We consider the task of generative modeling of 3D shapes (Yang et al., 2019). This task consists of learning to sample an arbitrary number of points $y$ from a surface $\Omega \subset \mathbb{R}^3$ that is described by a coarse $n$-point cloud $\mathbb{D} \in \Omega^n$. Diffusion models have recently proven to be very effective to solve this task, due to their expressivity and the high-quality of their output (Li et al., 2024b). We note $\Omega_f$ the set of points that a trained diffusion model $f$ can generate – in other words, the limit points of the denoising process.

This case is illustrated by Figure 1. The shape $\Omega$ is indicated by a dense cloud of red dots, the coarse dataset $\mathbb{D}$ by crosses, and the approximated shape $\Omega_f$ by a dense cloud of blue dots sampled from a trained diffusion model $f$. We can see in this example that the shape $\Omega$ features a rotational symmetry, which reduces the degree of freedom of the point cloud $\mathbb{D}$. If we know that the shape $\Omega$ is symmetric, a natural idea is to leverage this symmetry by using a rotation-equivariant architecture for the diffusion model $f$ (Hoogeboom et al., 2022). We show the output of a non-equivariant model on the left and of an equivariant model on the right.

We make the two following observations. First, the distribution learned by the non-equivariant neural network converges towards the empirical distribution $\frac{1}{|\mathbb{D}|} \sum_{y \in \mathbb{D}} \delta_y$, so that the approximate shape $\Omega_f$ coincides with the dataset $\mathbb{D}$. In other words, $\Omega_f = \mathbb{D}$. However, the equivariant model virtually

increases the diversity of the dataset $\mathbb{D}$ by the group of rotation $G$ to which it is equivariant. Thus, we find that $\Omega_f = G(\mathbb{D}) = \{g(z) | z \in \mathbb{D}, g \in G\}$.

It is clear from Figure 1 that the use of an equivariant network drastically improves the resolution of the approximate shape $\Omega_f$. In order to quantify this improvement, we introduce the representation gap

$$\mathcal{R}(\Omega, \Omega_f) = \int_\Omega \inf_{z \in \Omega_f} \|y - z\|_2^2 \, \mathrm{d}y \, , \tag{1}$$

a metric that measures how well $\Omega_f$ approximates the original shape $\Omega$. It is worth noticing that this metric is a natural generalization of the quantization error, which we recover when the set $\Omega_f$ is discrete (Graf & Luschgy, 2007).

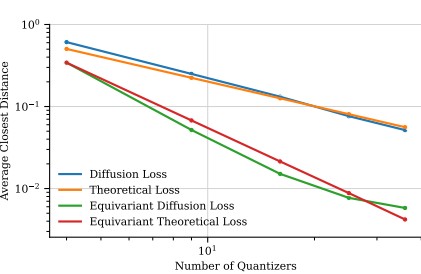

Figure 2: Log plot of the asymptotic evolution of the representation gap of a rotation-equivariant model and a non-equivariant model for a 2d-sphere surface. We observe a linear evolution, with slope $-1$ for the non-equivariant model and $-2$ for the equivariant model. The constant $J$ in Eq. 2 has been fitted for the theoretical curves.

The representation gap $\mathcal{R}(\Omega, \Omega_f)$ depends on how the dataset $\mathbb{D}$ has been sampled in $\Omega$. In practice, datasets $\mathbb{D}$ are collected as a best effort to cover the diversity of the task, modulo its known invariants. Correspondingly, we will assume that $\mathbb{D}$ is optimally sampled, *i.e.*, that it minimizes the risk in Equation 1. In this sense, our result can be seen as best-case scenarios.

Intuitively, a non equivariant model $f$ requires information about all the $d_\Omega = 2$ dimensions of the shape $\Omega$ in order to approximate it from the dataset $\mathbb{D}$ (as illustrated on the left of Figure 1). On the other hand, the equivariant model only needs information along the rotational axis, with dimension $1 = d_\Omega - 1$. More generally, for an arbitrary manifold $\Omega$ and symmetry group $G$, the equivariant model only needs information about the quotient space $\Omega/G$, with dimension $d_{\Omega/G}$. Indeed, the remaining dimensions are implicitly recovered by the virtual augmentation of the dataset, since $\Omega_f = G(\mathbb{D})$.

Concretely, let us note $n$ the size of an optimally sampled dataset $\mathbb{D}$, and $\mathcal{R}_n$ the representation gap of a model trained on $\mathbb{D}$. Then, we observe in Figure 2 that the representation gap scales as

$$\mathcal{R}_n \underset{n \to +\infty}{\sim} \frac{J}{n^{2/d}} \, , \tag{2}$$

where $d$ denotes either $d_\Omega$ in the case of a non-equivariant model or $d_{\Omega/G}$ in the case of an equivariant model. In this Equation, we can find a closed-form expression for the constant $J$, that depends only on the shape $\Omega$, the symmetry group $G$ and the Euclidean metric $\| \cdot \|_2^2$. Remarkably, the asymptotic evolution of the representation gap $\mathcal{R}_n(\Omega, \Omega_f)$ is governed by the single parameter $d$, that we name intrinsic dimension. This result characterizes precisely the advantage of the equivariant model over the non-equivariant one.

The purpose of the next Section is to prove formally these claims, and to extend our analysis to two more general settings – namely, supervised prediction and ambiguous prediction tasks.

## 4 THEORETICAL RESULTS

### 4.1 REPRESENTATION GAP FOR NON-CONDITIONAL DIFFUSION MODELS

We first consider the task of non-conditional diffusion models, and establish formally the claims of Section 3. We denote $\mathcal{Y} = \mathcal{R}^{d_\mathcal{Y}}$ the target space, of dimension $d_\mathcal{Y}$. We suppose that observations $y$ are structured and constrained to belong to a subset $\Omega \subset \mathcal{Y}$ of the ambient space. The set $\Omega$ models the world form which we draw observations. In particular, it captures its symmetries. Typically, we expected that these symmetries reduce the degree of freedom of the observations, so that $\Omega$ corresponds to a low-dimensional manifold of the ambient space $\mathcal{Y}$. This situation is commonly known as the manifold hypothesis (Bengio et al., 2013). In practice, we will consider that $\Omega$ is a Riemannian $d_\Omega$-manifold. We further suppose access to a dataset $\mathbb{D} \subset \Omega$ composed of $n$ observations drawn from $\Omega$. We consider neural networks $f_\theta$ in a parametric family $\mathcal{F}_\Theta \subset \mathcal{F}(\mathcal{Y} \times \mathbb{R}, \mathcal{Y})$. When there is no ambiguity, we will simply denote the neural networks by $f$.

We will focus on Denoising Diffusion Implicit Models (DDIM) diffusion models (Song et al., 2022). These models are trained to reverse a stochastic forward diffusion process that incrementally adds Gaussian noise to the data distribution while shrinking data points toward the origin. Noise addition is governed by a noise schedule $\alpha_t$, with $t \in [0, T]$. At each schedule step, the noised distribution can be written $\pi_t(y) = \sum_{z \in \mathbb{D}} \mathcal{N}(y|\sqrt{\alpha_t}z, (1 - \sqrt{\alpha_t})I)$. In particular, $\pi_0 = \frac{1}{|\mathbb{D}|} \sum_{z \in \mathbb{D}} \delta_z$ recovers the empirical data distribution and $\pi_T = \mathcal{N}(0, I)$ is an isotropic Gaussian distribution. In this context, diffusion models are trained to approximate the score function $s_t = \nabla \log \pi_t$ using the loss

$$\mathcal{L}(\theta) = \mathbb{E}_{t \sim \mathbb{U}[0,T], y_0 \sim \pi_0, \eta \sim \mathcal{N}(0,I)} \|f_\theta(\sqrt{\alpha_t}y_0 + \sqrt{1 - \alpha_t}\eta, t) - \eta\|_2^2 . \tag{3}$$

At sampling time, an initial point $y_T \sim \mathcal{N}(0, I)$ is sampled and then updated using the deterministic flow

$$\dot{y}_t = -\gamma_t(y_t + s_t(y_t)) , \tag{4}$$

where $t$ goes backward from $T$ to $0$. The output of the model corresponds to end points of this trajectory.

It can be shown that a diffusion model finding a global minimum of their training objective $\mathcal{L}$ – hence learning the true score function $s_t$ –, and following Equation 4 at sampling time, generate samples following the empirical distribution $\pi_0 = \frac{1}{|\mathbb{D}|} \sum_{z \in \mathbb{D}} \delta_z$ (Song & Ermon, 2019). In this case, the world representation $\Omega_f$ learned by the model $f$ is the training dataset $\mathbb{D}$ itself. Therefore, $\Omega_f = \mathbb{D}$ is a discrete approximation of the data manifold $\Omega$.

In practice, however, the neural network family $\mathcal{F}_\Theta$ has limited expressivity, which introduces biases when trying to estimate the score function $s_t$. Typically, the architecture of the neural network is chosen so that $f_\theta$ respects the symmetries of $\Omega$, and has therefore higher generalization capabilities. Remarkably, it is possible to show following Kamb & Ganguli (2025) that these architectural constraints virtually increase the diversity of the training dataset $\mathbb{D}$ via the symmetry group $G$ induced by the architecture, so that we have in effect $\Omega_f = G(\mathbb{D})$.

**Theorem 1** (Virtual augmentation of a dataset by a symmetry group). *See Proposition 3 in Appendix. Let $f$ denote a diffusion model equivariant under a symmetry group $G$ and minimizing the training objective in Equation 3 on a dataset $\mathbb{D}$. Then under mild assumptions on $G$, $\Omega$ and $\mathbb{D}$, the set of points that can be predicted by $f$ is $\Omega_f = G(\mathbb{D})$.*

*Proof.* The proof of Theorem 1 relies on the following observation: the score function $s_t$ at a point $y \in \mathcal{Y}$ can be written as an integral over the orbits $G(\mathbb{D})$ of the dataset $\mathbb{D}$:

$$s_t(y) = -\frac{1}{1 - \alpha_t} \int_{G(\mathbb{D})} (y - \sqrt{\alpha_t}z)W_t(z)\mathrm{d}z , \tag{5}$$

where each point $z \in G(\mathbb{D})$ is weighted by the distribution

$$W_t(z) = \frac{\mathcal{N}\left(y|\sqrt{\alpha_t}z, (1 - \alpha_t)I\right)}{\int_{G(\mathbb{D})} \mathcal{N}\left(y|\sqrt{\alpha_t}z', (1 - \alpha_t)I\right) \mathrm{d}z'} . \tag{6}$$

We can see that $W_t(y)$ acts as a softmax that peaks at the minimizer $y^* = \mathrm{argmin}_{z \in G(\mathbb{D})} \|y - z\|_2^2$ for small $t$. More precisely, we can use a Laplace approximation to show that $W_t(y)$ concentrate the probability mass around $y^*$ when $t \to 0$.

Under the hypothesis that $f$ minimizes the training objective in Equation 3, we can therefore write

$$f(y_t, t) = -\frac{1}{1 - \alpha_t} \int_{G(\mathbb{D})} (y_t - \sqrt{\alpha_t}z)W_t(z)\mathrm{d}z = \frac{1}{1 - \alpha_t}(y_t - y_t^*) + o\left(\frac{1}{1 - \alpha_t}\right) ,$$

which in turns implies $y_t - y_t^* \approx (1 - \alpha_t)f(y_t, t) \to 0$, and therefore $\lim_{t \to 0} y_t = \lim_{t \to 0} y_t^* \in G(\mathbb{D})$ (by properties of $G$). This proves $\Omega_f \subset G(\mathbb{D})$. The reverse inclusion is detailed in Appendix. $\square$

Using Theorem 1, we can characterize the asymptotic representation gap when the dataset size $n$ grows to infinity and $\mathbb{D}$ is optimally sampled. More precisely, we will note $\mathcal{R}_n = \inf_{\mathbb{D} \subset \Omega, |\mathbb{D}|=n} \mathcal{R}(\Omega, \Omega_f(\mathbb{D}))$ the representation gap of an optimally sampled dataset $\mathbb{D}$ of size $n$.

**Theorem 2** (Representation gap for non-conditional diffusion). *See Proposition 1, Proposition 2 and Proposition 4 in Appendix. Let $f$ denote a diffusion model equivariant under a symmetry group $G$ of isometries and minimizing the training objective in Equation 3 on a optimally sampled dataset $\mathbb{D}$ of size $n$. Suppose further that the orbits $G(z)$ have constant volume $|G|$ for each point $z \in \mathbb{D}$. Then under mild assumptions on $G$, $\mathbb{D}$ and $\Omega$, the representation gap satisfies*

$$\mathcal{R}_n \underset{n \to +\infty}{\sim} \frac{J_d |G| |\Omega/G|^{2/d}}{n^{2/d}} \ , \tag{7}$$

*where $\Omega/G$ denote the quotient space of $\Omega$ by the symmetry group $G$, $d = d_{\Omega/G}$ denote the dimension of $\Omega/G$, and $J_d$ is a constant that depends only on the quotient metric on $\Omega/G$ and the dimension $d$.*

*Proof.* We know by Theorem 1 that $\Omega_f = G(\mathbb{D})$. The idea is to apply factorize the integration over each orbit and recover the case of a discrete dataset $\mathbb{D}$. By standard properties of the orbits (see for instance Gallot et al. (1990)), and isometry of the elements of $G$

$$\mathcal{R}(\Omega, \Omega_f) = \int_\Omega \min_{z \in G(\mathbb{D})} \ell(y, z) \mathrm{d}y = |G| \int_{\Omega/G} \min_{z \in \mathbb{D}} \ell_{\Omega/G}(y, z) \mathrm{d}y \ .$$

We then conclude using a powerful result from quantization, Zador theorem, that characterizes the asymptotic behavior of the optimal quantization error (see for instance Theorem 2 in Gruber (2001) for a result on arbitrary manifolds). More precisely, we can show using this result that

$$\int_{\Omega/G} \min_{z \in \mathbb{D}} \ell_{\Omega/G}(y, z) \mathrm{d}y \underset{n \to +\infty}{\sim} \frac{J_d |G| |\Omega/G|^{2/d}}{n^{2/d}} \ .$$

$\square$

We recover Equation 2 by setting $J = J_d |G| |\Omega|^{2/d}$. Note that Theorem 2 provides an asymptotic equivalent of the representation gap, which is remarkable since most results about the generalization of neural network focuses on bounds (Zhang et al., 2021).

The constant $J_d$ has a closed-form expression which is unfortunately untractable in practice (see Theorem 8 in Appendix). However, for the Euclidean norm, it can be computed in simple cases ($J_1 = \frac{1}{12}$ and $J_2 = \frac{5}{18\sqrt{3}}$) and can be approximated for large $d$ by $J_d \sim \frac{d}{2\pi e}$ (Newman, 1982a; Pagès & Printems, 2003; Graf & Luschgy, 2007).

## 4.2 REPRESENTATION GAP FOR SUPERVISED PREDICTION

We now turn to the more general setting of supervised prediction. We denote $\mathcal{X} \subset \mathcal{R}^{d_\mathcal{X}}$ the input space, of dimension $d_\mathcal{X}$. Both $\Omega$ and $\mathbb{D}$ are now subsets of $\mathcal{X} \times \mathcal{Y}$. We note $\Omega_\mathcal{X} = \{x | (x, y) \in \Omega\}$ the projection of $\Omega$ to the input set $\mathcal{X}$ and $\Omega_\mathcal{Y} = \{y | (x, y) \in \Omega\}$ its projection to the target set $\mathcal{Y}$ (with similar definitions for $\mathbb{D}_\mathcal{X}$ and $\mathbb{D}_\mathcal{Y}$. Likewise we note $\Omega_x = \{y | (x, y) \in \Omega\}$ the data manifold conditioned by $x \in \Omega_\mathcal{X}$.

We consider ambiguous tasks, where each input $x \in \Omega_\mathcal{X}$ can be associated with potentially many targets $y \in \Omega_x$. We consider that $f$ captures the ambiguity of the task by generating several values. For instance, $f$ can be a conditional diffusion model providing a distribution over $\mathcal{Y}$ for each input $x \in \Omega_\mathcal{X}$ (Song & Ermon, 2019).

In this context the representation gap can be written

$$\mathcal{R}(\Omega, \Omega_f) = \int_{\Omega_\mathcal{X}} \int_{\Omega_x} \inf_{(x,y) \in \Omega_f} \|y_x - y\|_2^2 \mathrm{d}x \mathrm{d}y_x \ .$$

Note that we recover the empirical risk when the task is non-ambiguous (in this case, $\Omega_x$ is a singleton for each $x \in \Omega_\mathcal{X}$, f(x) takes a single value and we have $\mathcal{R}(\Omega, \Omega_f) = \int_{\Omega_\mathcal{X}} \|y_x - f(x)\|_2^2 \mathrm{d}x$).

First, we observe that Theorem 2 can be naturally extended to the setting where the input dataset $\Omega_\mathcal{X}$ is finite and covered by the dataset $\mathbb{D}_\mathcal{X}$ (see Proposition 5 in Appendix). However, the general case where $\Omega_\mathcal{X}$ is continuous requires some additional result on how the model $f$ behaves outside the input dataset $\mathbb{D}_\mathcal{X}$.

The recent literature about implicit regularization (Neyshabur et al., 2015) has shown that several popular training algorithms converge in fact toward minimal-norm interpolator of the training data (Zhang et al., 2021), especially in the over-parametrized regime (Allen-Zhu et al., 2019). Based on this work, we will consider that $f$ is a DDPM diffusion model which is smooth both regarding the conditioning $x$ and the noise input $y_t$. Therefore, it tends to project an input $(x, y_T)$ with initial noise $y_T$ toward a neighboring dataset point $(x, y) \in \mathbb{D}$. More precisely, we will suppose that there is a constant $L > 0$ so that if $z' \in B(z, L)$ for $z \in \mathbb{D}$, then $(x', y) \in \Omega_f$. Under this assumption, estimating the representation gap becomes close to the covering problem (i.e., finding an optimal covering of $\Omega$ with balls of constant radius), and we can derive the following bound.

**Theorem 3** (Conditional representation gap for ambiguous tasks). *Under mild assumptions on $\Omega$ and smoothness assumptions on the model $f$, the representation gap for a dataset $\mathbb{D}$ of size $n$ satisfies*

$$\mathcal{R}_n(\Omega, \Omega_f) \underset{n \to +\infty}{=} O\left(\frac{1}{n^{2/d_\Omega}}\right) . \tag{8}$$

*Proof.* Assume that $\mathbb{D}$ is an $\varepsilon$ covering of $\Omega$, for some $\varepsilon > 0$. Under the smoothness assumption, we have

$$\mathcal{R}(\Omega, \Omega_f) \leq \int_\Omega \min_{z' \in \mathbb{D}} \|z - z'\|_2^2 \mathrm{d}z \ \leq |\Omega|\varepsilon^2 , \tag{9}$$

so that the representation gap is tightly linked to the radius $\varepsilon$ of the covering.

Moreover, the size $N(\varepsilon)$ of the covering set $\mathbb{D}$ satisfies $N(\varepsilon) \leq 3^d \frac{|\Omega|}{|B|} n$ (see for instance Theorem 14.2 in Wu & Yang (2016)). Letting $\varepsilon = \frac{1}{n^{1/d}}$, $m = \left\lfloor 3^{-d} \frac{|\Omega|}{|B|} \right\rfloor$, and using that $\mathcal{R}_n$ is decreasing, we can conclude with the following observation:

$$R_n \leq R_{1/m^{1/d}} \leq |\Omega| \frac{1}{m^{2/d}} \leq \frac{|\Omega|}{\left(3^{-d} \frac{|\Omega|}{|B|} n\right)^{2/d}} = O\left(\frac{1}{n^{2/d}}\right) .$$

$\square$

If we focus on non-ambiguous prediction task, the data manifold $\Omega$ becomes a surface indexed by $\Omega_\mathcal{X}$. Under mild assumptions on the smoothness of $\Omega$, and assuming that $f$ is minimal-norm interpolator of $\mathbb{D}$ for the Total-Variation norm, we obtain a similar bound:

$$\mathcal{R}_n(\Omega, \Omega_f) = O(\frac{1}{n^{2/d}}) , \tag{10}$$

where $d = d_\Omega$. This result can also be extended to equivariant model, in which case $d = d_{\Omega/G}$. Details are given in the Appendix (see Proposition 6 and Proposition 7). Next, we validate experimentally the theoretical results established in Section 4.

## 5 EXPERIMENTAL RESULTS

**Datasets.** We conduct experiments on two synthetic dataset for non-conditional generative modeling, and one dataset for ambiguous prediction. They are illustrated in Figure 3 and Figure 1. We also use the MNIST dataset (Lecun et al., 1998).

- *Hypercube* corresponds to a $d_\Omega$-dimensional hypercube $\Omega = \left[-\frac{c}{2}, \frac{c}{2}\right]^{d_\Omega}$ of side $c$ embedded into a $d_\mathcal{Y}$ ambient space. This dataset features translation invariance over each of its dimensions.
- *Hypersphere* corresponds to a 2-dimensional hypersphere $\partial B(0, r)$ of radius $r$ embedded into a 3-dimensional ambient space. This dataset features many rotation-invariances (e.g. along axes $x$, $y$ and $z$).
- *Wave* corresponds to a 2-dimensional wave surface embedded into a 3-dimensional ambient space. The wave shape is obtain by concatenating two half-circle (along axes $x$ and $z$), and translating this curve along the $y \in [0, 1]$ segment. This dataset correspond to a conditional prediction task, where $x$ is the input and $(y, z)$ is the target. This dataset features translation invariance over the axis $y$.

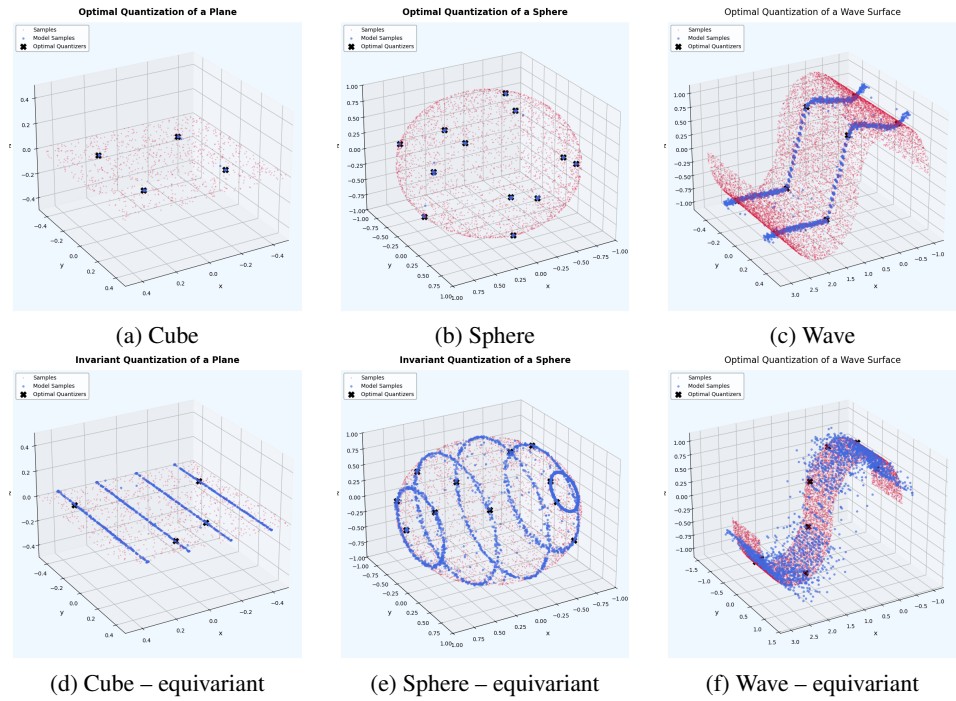

(a) Cube  (b) Sphere  (c) Wave

(d) Cube – equivariant  (e) Sphere – equivariant  (f) Wave – equivariant

Figure 3: Comparison of diffusion outputs across three datasets (cube, sphere, wave), with and without invariance constraints. We use the same legend as in Figure 1.

- *MNIST* (Lecun et al., 1998) is a dataset consisting 28x28 grayscale image of digit handrighting. The training dataset has size 60k and the test dataset has size 10k.

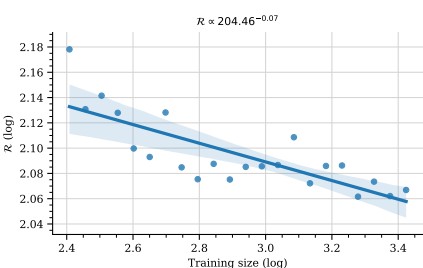

Figure 4: Log plot of the asymptotic evolution of the representation gap for the MNIST dataset.

**Architecture.** For the non-conditional task, we use a three-layer MLP (Rumelhart et al., 1986) with ReLU activation and 128 hidden units. For the conditional task, we use a 10-layer MLP with SiLU activation (Ramachandran et al., 2017), 128 hidden units, residual connections, and linear embedding for the conditioning. Translation or rotation equivariance is added on top of the corresponding architecture. For the MNIST experiment, we use a 2D U-Net backbone, implemented using the publicly available Hugging Face's Diffusers library (Von Platen et al., 2022).

**Training and optimization.** For the synthetic experiments, we use a DDIM diffusion model (Song et al., 2022), trained with a linear temperature schedule with $T = 100$ steps. We use the $\mathcal{L}_2$ loss defined on the ambient space $\mathcal{Y}$. The models are trained with the Adam optimizer (Kingma & Ba, 2017) for 50000 steps, with learning rate $\lambda = 10^{-3}$. All synthetic experiments are performed for 5 different seeds, and we report mean value and standard deviation. For the MNIST experiements, we use a DDPM diffusion model (Ho et al., 2020), trained during 2000 epochs, with the original temperature schedule and $T = 1000$ steps. This setup was sufficient for convergence.

**Metric.** In order to compute the representation gap, we sample 1000 point from the trained diffusion model, and 1000 points from the $\Omega$ (uniformly). We then compute the average minimum distance between these two cloud of points using Equation 1.

### 5.1 QUALITATIVE ANALYSIS

We can make two observations from Figure 1 and Figure 3. First, non-equivariant models converge toward the empirical distribution, so that $\Omega_f = \mathbb{D}$. Second, equivariant models converge towards

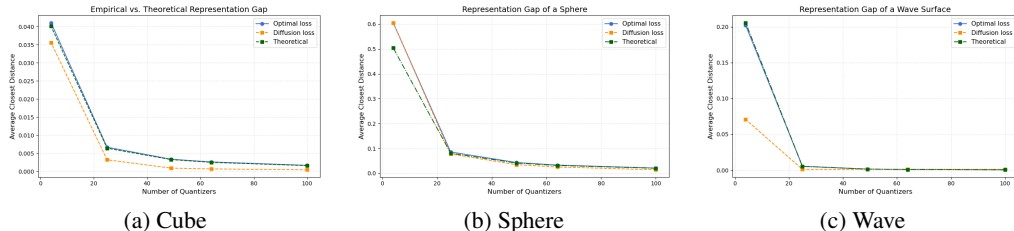

(a) Cube        (b) Sphere        (c) Wave

Figure 5: Asymptotic behavior of the representation gap across the three datasets of Figure 3. We plot the theoretical loss in Equation 7 (green), the representation gap $\mathcal{R}_n(\Omega, \mathbb{D})$ computed from the dataset points $\mathbb{D}$ (blue) and the empirical representation gap $\mathcal{R}_n(\Omega, \Omega_f)$ computed from a diffusion model $f$ trained on $\mathbb{D}$ (orange).

the empirical distribution virtually augmented by the invariance group $G$, so that $\Omega_f = G(\mathbb{D})$. This observation is confirmed across different shapes and number of points. It validates the claim of Proposition 1. This is a remarkable result, since the models $f$ are trained with the generic $\mathcal{L}_2$ loss and have no knowledge of the structure of the data manifold $\Omega$.

## 5.2 QUANTITATIVE ANALYSIS

In order to validate more precisely the formula in Proposition 2, we compute the asymptotic representation gap for different surfaces $\Omega$. The result of this experiment is given in Figure 5. For all datasets, the three curves follow the same asymptotic evolution and the difference between them are statistically insignificant. Note that conducting experiments on high dimension $d_\Omega$ is challenging, as the number of points $k^{d_\Omega}$ increases very fast and becomes quickly intractable. Moreover, using a lower dimension $d_\Omega$ is also challenging, since it makes the optimization problem harder (Hornik, 1991; Xu et al., 2025) . However it was possible to find a sweet spot between these two constraints.

## 5.3 MNIST EXPERIMENTS

Note that the MNIST dataset corresponds to the case where the input set $\Omega_{\mathcal{X}}$ is discrete, and covered by the dataset $\mathbb{D}_{\mathcal{X}}$. Therefore, the Proposition 5 in Appendix applies. In Figure 4, we plot the representation gap as a function of training dataset size. From the figure, we observe that the representation gap decreases linearly (in log-scale) as the training dataset size increases, which confirms the result of Proposition 2. By performing a linear regression, we obtain the relationship $\mathcal{R}_n(\Omega, \Omega_f) \propto 204.46^{-0.07n}$, and can therefore deduce that this task has an intrinsic dimension of $d \approx 14$. This is compatible with the ambient dimension of 784 point, and confirm that the task is relatively easy.

## 6 CONCLUSION

In the present work, we introduce a new metric – the representation gap–, that characterizes from a geometric point of view the generalization of neural networks. We provide a detailed asymptotic analysis of this representation gap in three important settings: non-conditional generative modeling, supervised prediction, and ambiguous task. We show that this representation gap is governed by a single parameter, the intrinsic dimension of the task. In particular, we show how standard machine learning techniques such as equivariant architecture reduces this intrinsic dimension, hence provably improving generalization. We validate our theoretical results and hypothesis on different carefully curated synthetic data and real-world data. We believe that intrinsic dimension could be leveraged to inform network architecture and training pipeline design in a principled manner. More generally, we argue that our present work introduces a new avenue for research on neural network generalization from a geometric perspective, through the lens of the representation gap. Indeed, the representation gap characterizes how, at test time, a trained neural network projects new inputs into the virtual manifold $\Omega_f$ that it learns from the training data $\mathbb{D}$ and from its invariants $G$. We believe this characterization could be the basis to study distribution shift at test time, novelty introduction (especially in the context of time-series forecasting), and more generally, the limits of statistical learning.

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

## A    INTRODUCTION

The Appendix is structured as follows. Section B introduces our notations, Section C introduces our main hypotheses and Section D describes main results from the literature on which our analysis relies. Then, Section E describes our results on non-conditional generative modeling and Section F describes our results on supervised prediction and ambiguous tasks.

## B    NOTATIONS

We consider supervised task, and denote $\mathcal{X} \subset \mathcal{R}^{d_{\mathcal{X}}}$ the set of input and $\mathcal{Y} \subset \mathcal{R}^{d_{\mathcal{Y}}}$ the set of targets. $d_{\mathcal{X}}$ and $d_{\mathcal{Y}}$ corresponds to the dimensions of these respective spaces. We suppose that observations $(x, y)$ are structured and constrained to belong to a subset $\Omega \subset \mathcal{X} \times \mathcal{Y}$ of the possible couplings. The set $\Omega$ models the world form which we draw observations. In particular, it captures its symmetries. Typically, we expected that these symmetries reduce the degree of freedom of the observations, so that $\Omega$ corresponds to a low-dimensional manifold of the ambient space $\mathcal{X} \times \mathcal{Y}$. This situation is usually referred to as the manifold hypothesis (Bengio et al., 2013). More precisely, we will consider that $\Omega$ is a $d$-Riemannian manifold (see for instance Lee (2006)).

We suppose access to a dataset $\mathbb{D} \subset \Omega$ composed of $n$ observations. We note $\mathbb{D}_x = \{y|(x, y) \in \mathbb{D})\}$ the targets' dataset $\mathbb{D}$ conditioned by a given $x$, and we will note $\mathbb{D}_{\mathcal{X}} = \{x|(x, y) \in \mathbb{D}\}$ (resp. $\mathbb{D}_{\mathcal{Y}}$) the set of inputs (resp. targets) appearing in $\mathbb{D}$. We will note $\Omega_x = y|(x, y) \in \Omega$ the set of observations conditioned by a given context $x \in \mathcal{X}$, and note $y_x \in \Omega_{\mathcal{Y}}$ the target corresponding to the input $x \in \Omega_{\mathcal{X}}$.

We consider neural networks $f_\theta$ in a parametric family $\mathcal{F}_\Theta \subset \mathcal{F}(\mathcal{X}, \mathcal{Y})$. When there is no ambiguity, we will simplify the notation and denote the neural networks by $f$.

If $G$ is a group, we will denote $G(y) = \{g(y)|g \in G\}$ the orbit of a single point $y \in \mathbb{D}$ under the group $G$, and $G(\mathbb{D}) = \cup_{y \in \mathbb{D}} G(y)$ the orbit of the dataset $\mathbb{D}$. A model $f$ is said to be equivariant under the group $G$ if for all $x \in \mathcal{X}$, we have $g(f(x)) = f(g(x))$. We will often consider that $G$ is a Lie group acting by isometries on the manifold $\Omega$. In particular, we can define the quotient manifold $\Omega/G$, the quotient map from $\pi : \Omega \to \Omega/G$ and the quotient metric on $\Omega/G$ induced from $\Omega$ (Lee, 2006).

We denote $\delta_x$ the Dirac distribution centered at a point $x \in \mathcal{X}$. We denote by $\Pi(E, F)$ the set of joint distributions over measurable sets $E$ and $F$, and we denote by $\pi_{\#1}$ and $\pi_{\#2}$ the marginals of a distribution $\pi \in \Pi(E, F)$. Let us denote $k_\varepsilon(a, b) = \exp(-\frac{\ell(a,b)}{\varepsilon})$ a Gaussian kernel. Let us denote $\mathcal{N}(\mu, \sigma^2)$ the Gaussian distribution and $\mathcal{N}(y|\mu, \sigma^2)$ the evaluation of its density function at a point $y$. Let us denote $\delta_y$ the Dirac distribution centered at a point $y$. Let us denote $\mathbb{1}[E]$ the indicative function of a set $E$. We denote by $\mathbb{P}$ a probability distribution. We denote the Total Variation (TV) semi-norm of a model $f$ by $TV(f) = \int_{\Omega_{\mathcal{X}}} \int_{\Omega_x} \sqrt{\|\nabla f(x)\|_2^2} \mathrm{d}x \mathrm{d}y$.

We denote $|E|$ the cardinal of a set $E$ when $E$ is finite. If $E$ is measurable, we de note $|E|$ its measure. If $E$ is a set $\overset{\circ}{E}$ denote its interior.

We denote $\ell$ a metric in $\mathcal{Y}$. If not indicated otherwise, $\ell$ will always correspond to the Euclidean distance $\ell(a, b) = \frac{1}{2}\|a - b\|_2^2$. We denote $d(y, E) = \min\{d(y, y')|y' \in G(\mathbb{D})\}$.

For $\varepsilon > 0$, we call $\varepsilon$-covering of $\Omega$ a set of balls $(B_k)_{k \in [\![1,n]\!]}$ of radius $\varepsilon$ such that $\Omega \subset \bigcup_{k \in [\![1,n]\!]} B_k$. We then define the covering number of $\Omega$ as the smallest integer $N(\varepsilon)$ such that there exists an $\varepsilon$-covering of $\Omega$. Likewise, for $\varepsilon > 0$, we call $\varepsilon$-packing of $\Omega$ a set of pairwise non-intersecting balls $(B_k)_{k \in [\![1,n]\!]}$ of radius $\varepsilon$ such that $\bigcup_{k \in [\![1,n]\!]} B_k \subset \Omega$. We then define the packing number of $\Omega$ as the largest integer $M(\varepsilon)$ such that there exists an $\varepsilon$-packing of $\Omega$.

The optimal quantization error, also called optimal quantization risk is defined by

$$\mathcal{R}_n(\mathbb{P}) = \inf_{z \in \mathcal{Y}^n} \int_{\mathcal{Y}} \min_{k \in [\![1,n]\!]} \|y - z_k\|^2 p(y) \mathrm{d}y \, ,$$

where $\mathbb{P}$ is a data distribution over $\mathcal{Y}$ admitting a density $p$.

We will focus on Denoising Diffusion Implicit Models (DDIM) diffusion models (Song et al., 2022). These models are trained to reverse a stochastic forward diffusion process that incrementally adds Gaussian noise to the data distribution while shrinking data points toward the origin. Noise addition is governed by a noise schedule $\alpha_t$, with $t \in [0, T]$. At each schedule step, the noised distribution can be written $\pi_t(y) = \sum_{z \in \mathbb{D}} \mathcal{N}(y|\sqrt{\alpha_t}z, (1 - \sqrt{\alpha_t})I)$. In particular, $\pi_0 = \frac{1}{|\mathbb{D}|} \sum_{z \in \mathbb{D}} \delta_z$ recovers the empirical data distribution and $\pi_T = \mathcal{N}(0, I)$ is an isotropic Gaussian distribution. In this context, the model $f_\theta : \mathcal{Y} \times \mathbb{R} \to \mathcal{Y}$ is trained to approximate the score function $s_t = \nabla \log \pi_t$ using the loss

$$\mathcal{L}(\theta) = \mathbb{E}_{t \sim \mathbb{U}[0,T], y_0 \sim \pi_0, \eta \sim \mathcal{N}(0,I)} \|f_\theta(\sqrt{\alpha_t}y_0 + \sqrt{1 - \alpha_t}\eta, t) - \eta\|_2^2 . \tag{11}$$

At sampling time, an initial point $y_T \sim \mathcal{N}(0, I)$ is sampled and then updated using the deterministic flow

$$\dot{y}_t = -\gamma_t(y_t + s_t(y_t)) , \tag{12}$$

where $t$ goes backward from $T$ to $0$. The output of the model corresponds to end points of this trajectory.

These equations can be generalized to the conditional case. In particular, the model $f_\theta : \mathcal{X} \times \mathcal{Y} \times \mathbb{R} \to \mathcal{Y}$ is trained using the loss

$$\mathcal{L}(\theta) = \mathbb{E}_{t \sim \mathbb{U}[0,T], (x_0,y_0) \sim \pi_0, \eta \sim \mathcal{N}(0,I)} \|f_\theta(x, \sqrt{\alpha_t}y_0 + \sqrt{1 - \alpha_t}\eta, t) - \eta\|_2^2 . \tag{13}$$

## C  HYPOTHESES

We will make repeated use of the following hypotheses.

**Assumption 4** (Optimal diffusion model). *The model $f$ is DDIM diffusion model minimizing the training objective defined in Equation 11.*

**Assumption 5** (Equivariance). *The model $f$ is equivariant under the group $G$, i.e. $f(g(x)) = g(f(x))$ for all $g \in G$ and $x \in \mathcal{X}$.*

**Assumption 6** (minimal-norm interpolator). *The model $f$ is a piecewise constant interpolator of the training data $\mathbb{D}$.*

**Assumption 7** (smooth covering model). *There is a constant $L > 0$ so that if $z' \in B(z, L)$ for $z \in \mathbb{D}$, then $(x', y) \in \Omega_f$.*

Note that the minimal-norm hypothesis 6 is met if $f$ is a minimal-norm interpolator of the training data $\mathbb{D}$ for the TV seminorm (Bredies & Vicente, 2019). Regularizing total variation has proved useful for a wide range of task, in particular in imaging applications (Huo et al., 2022; Jia et al., 2019), and has been for instance studied by the authors of Luo et al. (2025).

Likewise, the smooth covering hypothesis 7 is met by a conditional diffusion model $f$ if it is sufficiently smooth with respect to both its conditioning $x$ and its noisy input $y$. The smoothness of trained diffusion model has been studied both empirically (Guo et al., 2024) and theoretically (Liang et al., 2024a) by the recent literature, so that we believe that this hypothesis holds in practice.

## D  PREREQUISITE

We will use Zador's theorem (Zador, 1982), a powerful result on the asymptotic distribution of the centroids resulting from optimal quantization, which we recall below (see Graf et al. (2008), Equation 2.3, or Iacobelli (2016), Theorem 1.3, for a more general version).

**Theorem 8** (Zador theorem). *Let $\mathbb{P} = p \, \mathrm{d}y$ be a Lebesgue-dominated probability measure on a compact subset $\mathcal{Y}$ of $\mathbb{R}^d$. Define the optimal quantization risk*

$$\mathcal{R}_n(\mathbb{P}) = \inf_{z \in \mathcal{Y}^n} \int_\mathcal{Y} \min_{k \in [\![1,n]\!]} \|y - z_k\|_2^2 \, p(y) \mathrm{d}y ,$$

*and the asymptotic risk for the uniform distribution $J_d = \inf_n n^{2/d} \mathcal{R}_n(\mathcal{U}([0,1]^d))$. Then*

$$\lim_{n \to +\infty} n^{2/d} \mathcal{R}_n(\mathbb{P}) = J_d \left( \int_\mathcal{Y} p^{d/(d+2)} \mathrm{d}y \right)^{(d+2)/d} .$$

*In addition, if $f$ minimizes the risk $\mathcal{R}_n(\mathbb{P})$, then*

$$\frac{1}{n}\sum_{k=1}^{n}\delta_{z_k} \underset{n\to\infty}{\to} \frac{p^{d/(d+2)}}{\int_{\mathcal{Y}} p^{d/(d+2)}(y')\mathrm{d}y'}\mathrm{d}y\ .$$

The constant $J_d$ can be computed for simple cases ($J_1 = \frac{1}{12}$ and $J_2 = \frac{5}{18\sqrt{3}}$ (Newman, 1982b)) and can be approximated for large $d$ by $J_d \sim \frac{d}{2\pi e}$ (Pagès & Printems, 2003; Graf & Luschgy, 2007).

A generalization of Zador theorem to arbitrary manifolds has been proposed in Gruber (2001), that we report below (see Theorem 2 in this reference for a stronger result).

**Theorem 9** (Zador theorem on manifold). *Let $\|\cdot\|$ denote a norm on $\Omega$. Then there is a constant $J$ depending only on $\|\cdot\|$ such for all $J \subset \Omega$ compact and measurable with $|J| > 0$ and all $p : J \to \mathbb{R}^+$ continous, we have*

$$\inf_{z\in\mathcal{Y}^n}\int_J \min_{k\in[\![1,n]\!]}\|y-z_k\|^2 p(y)\mathrm{d}y \underset{n\to\infty}{\sim} J\left(\int_J p(u)^{\frac{d}{d+2}}\right)^{\frac{d+2}{d}}\frac{1}{n^{2/d}}\ . \tag{14}$$

# E    NON-CONDITIONAL TASKS

## E.1    MEMORIZING NETWORKS AND REPRESENTATION GAP

Let us first consider the case of a non-conditional prediction task. This setting corresponds to unconditional generative modeling, where the goal is to learn a probability distribution over $\Omega \subset \mathcal{Y}$ that captures its structure (*e.g.*, the support of the distribution is included in $\Omega$ and most common observations have higher probability).

Popular approaches for generative modeling include diffusion models (Ho et al., 2020; Song et al., 2022), Variational Auto Encoders (VAE) (Kingma & Welling, 2022), Generative Adversarial Networks (GAN) (Goodfellow et al., 2014) or normalizing flows (Rezende & Mohamed, 2016). Among them, diffusion models can be shown to converge toward the empirical distribution $\frac{1}{|\mathbb{D}|}\sum_{y\in\mathbb{D}}\delta_y$ when they minimize their training objective (Song & Ermon, 2019).

We will focus on this class of models hereafter. In this case, the empirical distribution corresponds to the world representation $\Omega_f$ learned by the model $f$, which can be seen as a discrete approximation of $\Omega$. We can compare this discrete word-representation $\Omega_f$ to $\Omega$ using the optimal quantization error

$$\mathcal{R}(\Omega,\Omega_f) = \int_\Omega \inf_{z\in\Omega_f}\ell(y,z)\mathrm{d}y\ . \tag{15}$$

This metric can be extended in the more general case where $\Omega_f$ may be continuous. We will refer to this distance as the representation gap. Note that quantity is notoriously difficult to study, even in discrete case (Graf & Luschgy, 2007). However, it becomes amenable to analysis in the asymptotic regime.

## E.2    REPRESENTATION GAP IN THE GENERAL CASE

Using this representation gap, we can characterize the difficulty of a task in terms of its sample efficiency.

**Proposition 1** (Representation gap). *Let us assume that $\Omega$ is Lebesgue-measurable with positive measure. Then, the optimal representation gap a model of a diffusion model $f$ minimizing its training objective 3 on a training dataset of size $n$ is*

$$\mathcal{R}(\Omega,\Omega_f)\underset{n\to+\infty}{\sim}\frac{J_d|\Omega|^{2/d}}{n^{2/d}}\ . \tag{16}$$

*Proof.* This is a corollary of Zador Theorem 8, in the particular case of a uniform distribution over $\Omega$. $\square$

This result is remarkable, since it provides an asymptotic equivalent of the representation gap as the dataset size $n$ grows to infinity. Most notably, the leading constant depends on the geometry of $\Omega$ only via its volume $|\Omega|$.

### E.3 Representation gap under the manifold hypothesis

It is possible to extend this result when $\Omega$ has null measure. This situation would typically arise under the manifold hypothesis. This hypothesis is interesting because it captures the structure of the observation world $\Omega$: even though the observation could a priori be an arbitrary point of $\mathcal{Y}$, it is in effect restricted to a low dimensional subspace $\Omega$.

**Proposition 2** (Representation gap under the manifold hypothesis). *Assume that $\Omega$ is a bounded Riemannian $d_\Omega$-manifold, and that $\ell$ is a norm on $\Omega$. Then the optimal representation gap of a diffusion model $f$ minimizing its training objective 3 on a training dataset $\mathbb{D}$ of size $n$ is*

$$\mathcal{R}(\Omega, \Omega_f) \underset{n \to +\infty}{\sim} \frac{J_{d_\Omega} |\Omega|^{2/d_\Omega}}{n^{2/d_\Omega}} . \tag{17}$$

*Proof.* This is a corollary of Theorem 2 in Gruber (2001). We satisfy the hypothesis of this Theorem, since the square function satisfies the growth condition and $\Omega$ is compact by hypothesis. We should only check that $J_{d_\Omega} |\Omega|^{2/d_\Omega}$ corresponds to the constant $J$ in the theorem. This is the case, since the constant does not depend on the integration set, and we can use $[0,1]^{d_\Omega}$ as long as it belongs to $\Omega$ (if not we can always use a scaling and translation of it that belongs to $\Omega$). □

This asymptotic evolution is similar to the general case described in Proposition 1, but leverages the structure of $\Omega$ via the lower dimension $d_\Omega$. Note that it is compatible with it in the case where $\Omega$ has positive measure in $\mathcal{Y}$. Again, it is remarkable that the leading constant depends on the geometry of $\Omega$ only via its volume $|\Omega|$. Moreover, it can be proved that the optimal data placement for $\mathbb{D}$ is uniformly distributed in $\Omega$ (cf. point 2.82 in Gruber (2001)).

### E.4 Representation gap for equivariant models

In practice, $\mathcal{F}_\Theta$ has limited expressivity, which introduces biases in the minimizer $f = \mathrm{argmin}_{\theta \in \Theta} \mathcal{L}(\theta)$. Typically, the architecture of the neural network is chosen so that $f_\theta$ respects the symmetries of $\Omega$, and has therefore higher generalization capabilities. Remarkably, the authors of Kamb & Ganguli (2025) have shown in the context of diffusion models that these architectural constraints increase the diversity of the dataset $\mathbb{D}$ via the symmetry group introduced by the architecture.

The following result is an extension of Theorem B.3 in Kamb & Ganguli (2025) to general symmetry groups $G$. More precisely, we will focus our attentions on Lie groups, which are a powerful way to define a large family of invariants that appear naturally in neural networks (Bronstein et al., 2021). They are also used in various fields such as physics, where they reflect the structure and symmetries of many physical systems (Gilmore, 2006; Georgi, 2000). This makes them particularly relevant for our purposes.

**Proposition 3** (Virtual augmentation of a dataset by a symmetry group). *Let us make the following assumptions*

*(i) $f$ is a trained diffusion model equivariant to $G$.*

*(ii) $G$ is a Lie group acting smoothly on the Riemannian manifold $\Omega$.*

*(iii) The distance $d(y, G(\mathbb{D}))$ is reached at a unique point $y^* = \Pi_{G(\mathbb{D})}(y) \in G(\mathbb{D})$ for all $y \in \mathcal{Y}$.*

*(iv) Let $y_t$ denote the denoising trajectory from the Gaussian distribution $\pi_T$, standard reverse diffusion process $\partial_t y_t = -\gamma_t(y_t + f(y_t, t))$. Assume that $y_t$ converge and $\partial_t y_t$ is bounded for each initial point $y_T$.*

*Then, the denoising trajectory ends at $y_0 \in G(\mathbb{D})$.*

*If we further assume each dataset point $z \in \mathbb{D}$ is a fixed point of the $f(\cdot, t)$ for all $t$, then each point $z \in G(\mathbb{D})$ is a limit point of the reverse diffusion process.*

Proposition 3 essentially states that under mild assumptions, an equivariant diffusion model $f$ will generate sample in the virtually augmented dataset $G(\mathbb{D})$. This is because the vision of the model $f$ is blurred due to its equivariance to $G$, so that it cannot distinguish points along the orbits $G(y)$ of the dataset points $y \in \mathbb{D}$.

The hypothesis $(i)$ states that the model $f$ is a global minimum of its training objective $\mathcal{L}$. The hypothesis $(ii)$ restricts our attention to Lie groups $G$, as discussed above. The point $(iii)$ avoids the degenerate case where the initial point $y$ is equidistant to a subset of the orbit of the dataset $G(\mathbb{D})$. Finally, the point $(iv)$ is a slightly relaxed form of a technical assumption introduced in Theorem B.3 of Kamb & Ganguli (2025). Finally, the fixed-point hypothesis captures the fact that each point $z \in \mathbb{D}$ is a local attractors of the score function, since the empirical distribution is discrete in our setting.

The proof of Proposition 3 relies on the following observation: the score function can be written as an integral over the orbits $G(z)$ of each data point $z \in \mathbb{D}$, where each point $z$ is weighted by the distribution

$$W_t(z) = \frac{\mathcal{N}\left(y|\sqrt{\alpha_t}z, (1-\alpha_t)I\right)}{\int_{G(\mathbb{D})} \mathcal{N}\left(y|\sqrt{\alpha_t}z', (1-\alpha_t)I\right) \mathrm{d}z'} \ . \tag{18}$$

In the case where the group $G$ is finite, we can see that $W_t(z)$ acts as a softmax that peaks when $z^*$ as $t \to 0$. In the more general case where $G$ is not finite, we can use a Laplace approximation to show that $W_t(z)$ concentrate the probability mass around the minimizer $z^*$ when $t \to 0$. Therefore, the denoising trajectory is attracted toward the orbit $G(\mathbb{D})$.

**Lemma 1** (Laplace approximation). *Let $G$ denote a Lie group acting smoothly on $\Omega$, $\alpha_t$ a continuous positive noise schedule satisfying $\alpha_t \to_{t\to 0} 1$, $y \in \mathcal{Y}$ an arbitrary point, $d$ the dimension of $G(\mathbb{D})$, and $h$ a bounded continuous non-negative function on $G(\mathbb{D})$. Assume that $y$ has a unique closest point $y^* \in G(\overset{\circ}{\mathbb{D}})$, the interior of the orbit. Define $\beta_t = 2\frac{1-\alpha_t}{\alpha_t}$ a temperature scaling. Then, we have*

$$\int_{G(\mathbb{D})} h(z)\mathcal{N}\left(y|\sqrt{\alpha_t}z, (1-\alpha_t)I\right) \mathrm{d}z \underset{t\to 0}{=} h(y^*)\, e^{-\|y^*-y\|^2/\beta_t}\, (2\pi\beta_t)^{d/2} \tag{19}$$

$$+ o\left(e^{-\|y^*-y\|^2/\beta_t}\, \beta_t^{d/2}\right) \ .$$

*Proof.* Let us denote by $I(t) = \int_{G(\mathbb{D})} h(y)\mathcal{N}\left(y|\sqrt{\alpha_t}z, (1-\alpha_t)I\right) \mathrm{d}z$ the left term in Equation 19. Informally, the proof of Lemma 1 then relies on the two following approximations:

$$I(t) = \int_{G(\mathbb{D})} h(z)e^{-\|z-\frac{y}{\alpha_t}\|^2/\beta_t} \mathrm{d}z \approx \int_{G(\mathbb{D})} h(z)e^{\|z-y\|^2/\beta_t} \mathrm{d}z \approx h(y^*)e^{-\|y^*-y\|^2/\beta_t}(2\pi\beta_t)^{d/2} \ .$$

The first approximation comes from integrating $\|z - \frac{y}{\alpha_t}\|^2 = \|z - y\|^2 + O(\beta_t)$ over the orbit $G(\mathbb{D})$, and the second approximation is an extension of Laplace approximation on measurable subsets of $\mathbb{R}^d$. It expresses that the Gaussian kernel $e^{\|z-y\|^2/\beta_t}$ concentrates mass at the minimizer $y^*$, with a curvature term $(2\pi\beta_t)^{d/2}$.

Let us now prove these two approximations. First observe that

$$\|z - \frac{y}{\alpha_t}\|^2 - \|y^* - \frac{y}{\alpha_t}\|^2 = \|z - y\|^2 - \|y^* - y\|^2 + 2\frac{\sqrt{\alpha_t}-1}{\sqrt{\alpha_t}}\langle z - y^*|y\rangle,$$

so that by exponentiation and integration, we have

$$\int_{G(\mathbb{D})} h(z)e^{-\|z-\frac{y}{\alpha_t}\|^2/\beta_t} \mathrm{d}z = e^{-\|y^*-\frac{y}{\alpha_t}\|^2/\beta_t} \underbrace{\int_{G(\mathbb{D})} h(z)e^{\frac{\sqrt{\alpha_t}}{2(1+\sqrt{\alpha_t})}\langle y^*-z|y\rangle}\, e^{(\|y^*-y\|^2-\|z-y\|^2)/\beta_t} \mathrm{d}z}_{J(t)} \ .$$

The noise schedule $\alpha_t$ is bounded in $[0, 1]$, so that $e^{-|\langle y^*-z|y\rangle|} \le e^{\frac{\sqrt{\alpha_t}}{2(1+\sqrt{\alpha_t})}\langle y^*-z|y\rangle} \le e^{|\langle y^*-z|y\rangle|}$. Let us define

$$J_-(t) = \int_{G(\mathbb{D})} h(z)\, e^{-|\langle y^*-z|y\rangle|}\, e^{(\|y^*-y\|^2-\|z-y\|^2)/\beta_t} \mathrm{d}z \ ,$$

a lower bound of $J(t)$.

Then we can apply Corollary 3.4 in Kirwin (2010) to $J(t)$ in order to obtain that $J_-(t) \underset{t\to 0}{=} h(y^*)(2\pi\beta_t)^{d/2} + o(\beta_t^{d/2})$. Indeed, the conditions of this Corollary are met (modulo a change

of variable), since $G(\mathbb{D})$ is a measurable set which contains $y^*$ as an interior point, $z \mapsto \|y^* - y\|^2 - \|z - y\|^2$ is twice differentiable and attains its unique minimum value of 0 at $y^*$, $z \mapsto h(z)e^{-|\langle y^* - z|y \rangle|}$ is a continuous function on $G(\mathbb{D})$ evaluating at $h(y^*)$ on $y^*$, and $1/\beta_t \underset{t \to 0}{\to} +\infty$.

Likewise, we can also prove that

$$J_+(t) = \int_{G(\mathbb{D})} h(z)\, e^{|\langle y^* - z|y \rangle|}\, e^{(\|y^* - y\|^2 - \|z - y\|^2)/\beta_t} \mathrm{d}z \underset{t \to 0}{=} h(y^*)(2\pi\beta_t)^{d/2} + o(\beta_t^{d/2}) \ .$$

Therefore, we deduce by squeezing that $J(t) \underset{t \to 0}{=} h(y^*)(2\pi\beta_t)^{d/2} + o(\beta_t^{d/2})$, and we can conclude

$$I(t) = e^{-\|y^* - \frac{y}{\alpha_t}\|^2/\beta_t} J(t) \underset{t \to 0}{=} h(y^*)\, e^{-\|y^* - y\|^2/\beta_t}\, (2\pi\beta_t)^{d/2} + o\left( e^{-\|y^* - y\|^2/\beta_t}\, \beta_t^{d/2} \right) \ .$$

$\square$

We can now prove Proposition 3.

*Proof of Proposition 3.* By theorem B.3 in Kamb & Ganguli (2025), the score function by the model $f$ can be written

$$f(y_t, t) = -\frac{1}{1 - \alpha_t} \frac{\int_{G(\mathbb{D})} (y - \sqrt{\alpha_t}z)\mathcal{N}(y|\sqrt{\alpha_t}z, (1 - \alpha_t)I)\mathrm{d}z}{\int_{G(\mathbb{D})} \mathcal{N}(y|\sqrt{\alpha_t}z, (1 - \alpha_t)I)\mathrm{d}z} = \frac{1}{1 - \alpha_t}(y_t - y_t^*) + o\left(\frac{1}{1 - \alpha_t}\right), \tag{20}$$

where the second equality is a corollary of Lemma 1 to be justified later. Then, hypothesis $(iv)$ implies that $\gamma_t f(y_t, t) = \partial_t y_t + \gamma_t y_t$ is bounded, which in turn implies $y_t - y_t^* = (1 - \alpha_t)f(y_t, t) \to 0$. Since $y_t^* \in G(\mathbb{D})$, which is compact (by hypothesis $(ii)$ and property of Lie groups), and $y_t$ converge (by hypothesis $(iv)$), then $y_t^*$ converge and $\lim_{t \to 0} y_t = \lim_{t \to 0} y_t^* \in G(\mathbb{D})$.

Therefore, we only need prove the approximation in Equation 20. Noting $d$ the dimension of $G(\mathbb{D})$, $y_t^*$ the unique minimizer of $d(y_t, G(\mathbb{D}))$ (by hypothesis $(iii)$), and $I(t) = \int_{G(\mathbb{D})} (y - \sqrt{\alpha_t}z)\mathcal{N}(y|\sqrt{\alpha_t}z, (1 - \alpha_t)I)\mathrm{d}z$, we can write the following.

$$I(t) - (y_t - \sqrt{\alpha_t}y_t^*)(2\pi\beta_t)^{d/2} = \int_{G(\mathbb{D})} (y - \sqrt{\alpha_t}z)\mathcal{N}(y|\sqrt{\alpha_t}z, (1 - \alpha_t)I)\mathrm{d}z$$

$$- \int_{G(\mathbb{D})} (y - \sqrt{\alpha_t}y^*)\mathcal{N}(y|\sqrt{\alpha_t}z, (1 - \alpha_t)I)\mathrm{d}z$$

$$= \sqrt{\alpha_t} \int_{G(\mathbb{D})} (y^* - z)\mathcal{N}(y|\sqrt{\alpha_t}z, (1 - \alpha_t)I)\mathrm{d}z$$

$$\|I(t) - (y_t - \sqrt{\alpha_t}y_t^*)(2\pi\beta_t)^{d/2}\| \leq \sqrt{\alpha_t} \int_{G(\mathbb{D})} \|y^* - z\|\mathcal{N}(y|\sqrt{\alpha_t}z, (1 - \alpha_t)I)\mathrm{d}z$$

The function $z \mapsto \|y^* - z\|$ is bounded, continuous and non-negative on $G(\mathbb{D})$. Moreover, $z$ so that the conditions of Lemma 1 are met. Therefore, we deduce by bounding that $I(t) - (y_t - \sqrt{\alpha_t}y_t^*)(2\pi\beta_t)^{d/2} = o(\beta_t^{d/2})$, which entails $I(t) = (y_t - y_t^*)(2\pi\beta_t)^{d/2} + o(\beta_t^{d/2})$.

On the other side, we also deduce from Lemma 1 that $\int_{G(\mathbb{D})} \mathcal{N}(y|\sqrt{\alpha_t}z, (1 - \alpha_t)I)\mathrm{d}z = (2\pi\beta_t)^{d/2} + o(\beta_t^{d/2})$. Therefore, we have

$$f(y_t, t) = \frac{1}{1 - \alpha_t} \frac{(y_t - y_t^*)(2\pi\beta_t)^{d/2} + o(\beta_t^{d/2})}{(2\pi\beta_t)^{d/2} + o(\beta_t^{d/2})} = \frac{1}{1 - \alpha_t}(y_t - y_t^*) + o\left(\frac{1}{1 - \alpha_t}\right) \ .$$

This shows that $\Omega_f \subset G(\mathbb{D})$. For the reverse inclusion, we will use the assumption that each point $z \in \mathbb{D}$ is a fixed point of the model $f$. More precisely, assume that $y_t = g(z) \in \mathbb{D}$ with $g \in G$ and $z \in \mathbb{D}$. Then $\partial_t y_t = -\gamma_t(g(z) - f(g(z), t)) = -\gamma_T g(z - f(z, t)) = 0$ by equivariance of $f$ and by the fixed point hypothesis. Therefore, a trajectory starting at $y_T \in G(\mathbb{D})$ stays at $y_T$, which is hence a limit point.

This establishes $\Omega_f = G(\mathbb{D})$ and concludes the proof of Proposition 3. $\square$

Proposition 3 established that an equivariant diffusion model $f$ generated samples in $G(\mathbb{D})$. Therefore, we can identify its world representation $\Omega_f$ with $G(\mathbb{D})$. If the symmetry group $G$ enforced by the architecture is aligned with the symmetries of the world $\Omega$, then we can further improve the sample efficiency of $\mathbb{D}$.

**Proposition 4** (Representation gap for an equivariant function). *Assume that $\Omega$ is a bounded Riemannian $d_\Omega$-manifold, and $f$ is a diffusion model minimizing its training objective 3 on a training dataset $\mathbb{D}$ of size $n$. Assume further that $f$ is equivariant over a Lie group $G$ of isometries acting smoothly on $\Omega$, and the orbits $G(y)$ have the same Riemannian volume $|G|$ for each point $y \in \mathbb{D}$. Denote $d_{\Omega/G}$ the dimension of the quotient space $\Omega/G$. Then the representation gap of $f$ is*

$$\mathcal{R}(\Omega, \Omega_f) \underset{n\to+\infty}{\sim} \frac{J_{d_{\Omega/G}} |G| |\Omega/G|^{2/d_{\Omega/G}}}{n^{2/d_{\Omega/G}}} , \tag{21}$$

*where $J_{d_{\Omega/G}}$ uses the quotient metric $\ell_{\Omega/G}(\pi(a), \pi(b))$ and $\pi$ is the quotient map from $\Omega$ to $\Omega/G$.*

*Proof.* The idea is to apply Fubini theorem to factorize the integration over each orbit. We have from Proposition 3 that $\Omega_f = G(\mathbb{D})$. Therefore, by standard properties of the orbits (see for instance Gallot et al. (1990)) and using the isometry of the elements of $G$, we obtain

$$\mathcal{R}(\Omega, \Omega_f) = \int_\Omega \min_{z \in G(\mathbb{D})} \ell(y, z) \mathrm{d}y$$

$$= \int_{\Omega/G} \int_{\pi^{-1}(y)} \min_{z \in G(\mathbb{D})} \ell_{\Omega/G}(\pi(y'), \pi(z)) \mathrm{d}y \mathrm{d}y'$$

$$= \int_{\Omega/G} \int_{\pi^{-1}(y)} \min_{z \in \mathbb{D}} \ell_{\Omega/G}(y, z) \mathrm{d}y \mathrm{d}y'$$

$$= |G| \int_{\Omega/G} \min_{z \in \mathbb{D}} \ell_{\Omega/G}(y, z) \mathrm{d}y .$$

Therefore, we are in the setting of Proposition 2, since $\Omega/G$ is a manifold and $\ell_{\Omega/G}$ is a norm on $\Omega/G$ and $\mathrm{d}y$ is a Riemannian metric on $\Omega/G$. We can then conclude

$$\mathcal{R}(\Omega, \Omega_f) \underset{n\to+\infty}{\sim} \frac{J_{d_{\Omega/G}} |G| |\Omega|^{2/d_{\Omega/G}}}{n^{2/d_{\Omega/G}}} . \tag{22}$$

$\square$

Proposition 4 again features an asymptotic evolution similar to the general case described in Proposition 1 and the case of a manifold structure described in Proposition 2. In particular, we recover these formulas respectively when the group $G$ contains only the identity, and when the observation world $\Omega$ has positive measure.

# F  CONDITIONAL TASKS

## F.1  DISCRETE-CLASS CONDITIONING

We now extend these results to the more general case of conditional tasks. Both $\Omega$ and $\mathbb{D}$ are subsets of $\mathcal{X} \times \mathcal{Y}$. Let us first focus on the case where $\Omega_\mathcal{X}$ is finite and covered by the input dataset $\mathbb{D}_\mathcal{X}$. It is clear that for each input $x \in \mathbb{D}_\mathcal{X}$, the Propositions 1, 2 and 4 apply to the conditional dataset $\mathbb{D}_x$ and the conditional manifold $\Omega_x$. We summarize this observation in the following Proposition.

**Proposition 5** (Representation gap for discrete-class conditioning). *Assume that $\Omega_\mathcal{X}$ is finite, that we have $\mathbb{D}_\mathcal{X} = \Omega_\mathcal{X}$, and that $\Omega_x$ is a bounded Riemannian $d_\Omega$-manifold for each $x \in \Omega_\mathcal{X}$. Let $f$ denote a diffusion model minimizing its training objective 13 on a training dataset $\mathbb{D}$ of size $n$. Assume further that $f$ is equivariant over a Lie group $G$ of isometries acting smoothly on $\Omega_x$ for each $x \in \mathcal{X}$, and the orbits $G(y)$ have the same Riemannian volume $|G|$ for each point $(x, y) \in \mathbb{D}$, and that the . Noting $d_{\Omega/G}$ the common dimension of the quotient manifolds $\Omega_x/G$, the representation gap of $f$ can be written*

$$\mathcal{R}(\Omega, \Omega_f) \underset{n\to+\infty}{\sim} \frac{|G| \sum_{x \in \mathcal{X}} J_x |\Omega_x/G|^{2/d_{\Omega/G}}}{n^{2/d_{\Omega/G}}} , \tag{23}$$

*where $J_x$ uses the quotient metric $\ell_{\Omega_x/G}(\pi_x(a), \pi_x(b))$ and $\pi$ is the quotient map from $\Omega_x$ to $\Omega_x/G$.*

*Proof.* Application of Proposition 4 to each conditional dataset $\mathbb{D}_x$ the conditional manifold $\Omega_x$. □

Note that if the conditional manifolds $\Omega_x$ have different dimension $d_x$ for each $x \in \Omega_{\mathcal{X}}$, the representation gap is determined by the conditional manifolds with the highest dimension. In particular, the intrinsic dimension becomes $d = \max_{x \in \Omega_{\mathcal{X}}} d_x$.

### F.2 NON-AMBIGUOUS TASKS AND MINIMAL-NORM INTERPOLATORS

We now turn to the case where $\Omega_{\mathcal{X}}$ is continuous. Clearly, we require some result on how $f$ behaves outside the training data $\mathbb{D}_{\mathcal{X}}$.

We will first restrict our attention to non-ambiguous tasks: each input $x \in \Omega_{\mathcal{X}}$ is associated to a unique target $y_x \in \Omega_{\mathcal{Y}}$. Therefore, the observation world $\Omega$ can be see as a curve indexed by $\Omega_{\mathcal{X}}$. In particular, its dimension is $d_\Omega = d_{\Omega_{\mathcal{X}}} + 1$, no matter what is the dimension of $\mathcal{Y}$. We further consider that the model $f$ generates a single output $f(x)$ for each input $x \in \Omega_{\mathcal{X}}$, so that its representation $\Omega_f$ can also be seen as a curve indexed by $\Omega_{\mathcal{X}}$. In this context, the conditional representation gap can be defined by

$$\mathcal{R}(\Omega, \Omega_f) = \int_{\Omega_{\mathcal{X}}} \ell(y_x, f(x)) \mathrm{d}x \tag{24}$$

For the purpose of our analysis, we will rely on the recent literature about implicit regularization (Neyshabur et al., 2015). Indeed, several training algorithm have been shown to converge toward minimal-norm interpolator of the training data (Zhang et al., 2021), especially in the over-parametrized regime (Allen-Zhu et al., 2019). Examples have been given for the $\mathcal{L}_1$ norm (Liang & Sur, 2022; Gunasekar et al., 2018), the $\mathcal{L}_2$ norm (Liang & Rakhlin, 2018; Mei & Montanari, 2022; Hastie et al., 2022) or the Sobolev seminorm (Ma & Ying, 2021). In the case of diffusion model, a form of mode interpolation has been shown (Bonnaire et al., 2025).

In order to keep the problem tractable, we will focus on the total variation norm. However, For this norm, it has been shown in some settings that minimal-norm interpolator are piecewise constant functions (Bredies & Vicente, 2019). Basing ourselves on this observation, we will introduce the minimal-norm assumption 6 for the remaining of this Section.

### F.3 CONDITIONAL REPRESENTATION GAP UNDER THE MANIFOLD HYPOTHESIS

We now study how to generalize the result of Proposition 2 to the conditional setting. It is unclear weather we can derive a clean asymptotic equivalent of the representation gap in this case, since the geometry of $\Omega$ become critical due to the coupling between input and target. However, the next Proposition introduce an upper bound that follow the form introduced in 1, 2 and 4.

**Proposition 6** (Conditional representation gap under the manifold hypothesis). *Assume that $\Omega$ is a bounded Riemannian $d_\Omega$-manifold, and that $\ell$ is a norm on $\Omega$. Then the representation gap of the minimal-norm interpolator $f$ (assumption 6) of a dataset $\mathbb{D}$ of size $n$ satisfies*

$$\mathcal{R}(\Omega, \Omega_f) \underset{n \to +\infty}{=} O\left(\frac{1}{n^{2/(d_\Omega - 1)}}\right) . \tag{25}$$

*Proof.* Let us denote $\Omega = \{(x, \omega(x)) | x \in \Omega_{\mathcal{X}}\}$, and $\|\omega\|_\infty$ the norm of the gradient of $x \mapsto \omega(x)$. Under the assumption 6 that $f$ is a minimal-norm interpolator, we have that $f(x) = \omega(z_x)$, for $z_x = \mathrm{argmin}_{z \in \mathbb{D}} \ell(x, z)$. Therefore, we can write

$$\mathcal{R}(\Omega, \Omega_f) = \int_{\Omega_{\mathcal{X}}} \ell(\omega(x), f(x)) \mathrm{d}x$$

$$= \int_{\Omega_{\mathcal{X}}} \ell(\omega(x), \omega(z_x))$$

$$\leq \|\omega\|_\infty \int_{\Omega_{\mathcal{X}}} \ell(x, z_x)$$

$$= \|\omega\|_\infty \int_{\Omega_{\mathcal{X}}} \min_{z \in \mathbb{D}_{\mathcal{X}}} \ell(x, z) .$$

Using Proposition 4, we know that $\int_{\Omega_{\mathcal{X}}} \min_{z \in \mathbb{D}_{\mathcal{X}}} \ell(x, z) \underset{n \to +\infty}{\sim} \frac{J_{d_{\Omega_{\mathcal{X}}}} |\Omega|^{2/d_{\Omega_{\mathcal{X}}}}}{n^{2/d_{\Omega_{\mathcal{X}}}}}$ for optimally placed $z \in \mathbb{D}_{\mathcal{X}}$. We can therefore deduce

$$\mathcal{R}(\Omega, \Omega_f) \underset{n \to +\infty}{=} O\left(\frac{1}{n^{2/d_{\Omega_{\mathcal{X}}}}}\right) .$$

$\square$

### F.4 CONDITIONAL REPRESENTATION GAP FOR EQUIVARIANT MODEL

It is interesting to extend the result of Proposition 6 to the case where the model $f$ is equivariant to a set of symmetries $G$. In the non-conditional case (Proposition 4), it was the target space of the dataset $\mathbb{D}_{\mathcal{Y}}$ that was virtually augmented by the group $G$. In this case however, it is the input space of the dataset $\mathbb{D}_{\mathcal{X}}$ that is virtually augmented by $G$, as we will show below. As a consequence, we can derive a tighter upper bound leveraging the dimension $d_{\Omega_{\mathcal{X}}/G}$ of the quotient space $\Omega_{\mathcal{X}}/G$.

**Proposition 7** (Conditional representation gap of an equivariant function). *Assume that $\Omega$ is a bounded Riemannian $d_\Omega$-manifold, and that $\ell$ is a norm on $\Omega$. Let us further assume that $f$ is equivariant under a Lie group $G$ acting smoothly, freely and isometrically on $\Omega_{\mathcal{X}}$, and the orbits $G(y)$ have the same Riemannian volume $|G|$ for each point $y \in \mathbb{D}$. Denote by $d_{\Omega_{\mathcal{X}}/G}$ the dimension of the quotient space $\Omega_{\mathcal{X}}/G$. Then the representation gap of the minimal-norm interpolator $f$ of a dataset $\mathbb{D}$ of size $n$ (assumption 6) satisfies*

$$\mathcal{R}(\Omega, \Omega_f) \underset{n \to +\infty}{=} O\left(\frac{1}{n^{2/d_{\Omega_{\mathcal{X}}/G}}}\right) . \tag{26}$$

*Proof.* Under the assumption that $f$ is a minimal-norm interpolator equivariant to $G$ (assumptions 6 and 5), we have that $f(x) = \omega(z_x^G)$, for $z_x^G = \operatorname{argmin}_{z \in G(\mathbb{D}_{\mathcal{X}})} \ell(x, z)$ (the minimum is reached by the properties of Lie groups). We note $z_x = \operatorname{argmin}_{z \in \mathbb{D}_{\mathcal{X}}} \ell(x, z)$ as in the proof of Proposition 4, and $\pi$ the quotient map from $\Omega_{\mathcal{X}}$ to $\Omega_{\mathcal{X}}/G$. By using the isometry of the elements of G, factorizing the integration over each orbit, and noting $\pi$ the quotient map from $\Omega_{\mathcal{X}}$ to $\Omega_{\mathcal{X}}/G$, we can write

$$\mathcal{R}(\Omega, \Omega_f) = \int_{\Omega_{\mathcal{X}}} \ell(\omega(x), f(x)) \mathrm{d}x$$
$$= \int_{\Omega_{\mathcal{X}}/G} \int_{\pi^{-1}(x)} \ell_{\Omega_{\mathcal{X}}/G}(\omega(\pi(x')), \omega(\pi(z_{x'}^G))) \mathrm{d}x \mathrm{d}x'$$
$$= \int_{\Omega_{\mathcal{X}}/G} \int_{\pi^{-1}(x)} \ell_{\Omega_{\mathcal{X}}/G}(\omega(x), \omega(z_x))) \mathrm{d}x \mathrm{d}x'$$
$$= |G| \int_{\Omega_{\mathcal{X}}/G} \ell_{\Omega_{\mathcal{X}}/G}(\omega(x), \omega(z_x))) \mathrm{d}x .$$

Then, noting $\|\omega\|_\infty$ the norm of the gradient of $x \mapsto \omega(x)$ restricted to the manifold $\Omega_{\mathcal{X}}/G$, we know from the proof of Proposition 6 that

$$\int_{\Omega_{\mathcal{X}}/G} \ell_{\Omega_{\mathcal{X}}/G}(\omega(x), \omega(z_x))) \mathrm{d}x \leq \|\omega\|_\infty \int_{\Omega_{\mathcal{X}}/G} \min_{z \in \mathbb{D}_{\mathcal{X}}} \ell_{\Omega_{\mathcal{X}}/G}(x, z) .$$

Therefore, we are in the setting of Proposition 2, since $\Omega_{\mathcal{X}}/G$ is a manifold and $\ell_{\Omega_{\mathcal{X}}/G}$ is a norm on $\Omega/G$ and $\mathrm{d}y$ is a Riemannian metric on $\Omega_{\mathcal{X}}/G$. We can deduce

$$\int_{\Omega_{\mathcal{X}}/G} \min_{z \in \mathbb{D}_{\mathcal{X}}} \ell_{\Omega_{\mathcal{X}}/G}(x, z) \underset{n \to +\infty}{\sim} \frac{J_{d_{\Omega_{\mathcal{X}}/G}} |G| |\Omega_{\mathcal{X}}|^{2/d_{\Omega_{\mathcal{X}}/G}}}{n^{2/d_{\Omega_{\mathcal{X}}/G}}} . \tag{27}$$

Combining these result, we deduce

$$\mathcal{R}(\Omega, \Omega_f) \underset{n \to +\infty}{=} O\left(\frac{1}{n^{2/d_{\Omega_{\mathcal{X}}/G}}}\right) . \tag{28}$$

$\square$

### F.5 DISCUSSION ABOUT AMBIGUOUS TASKS

We now turn our attention to the most general case of ambiguous conditional prediction tasks. Both $\Omega$ and $\mathbb{D}$ are still subsets of $\mathcal{X} \times \mathcal{Y}$. However, each input $x \in \Omega_{\mathcal{X}}$ is now associated with potentially many targets $y \in \Omega_{\mathcal{Y}}$. As a consequence, the observation world $\Omega$ cannot be seen as curve indexed by $\Omega_{\mathcal{X}}$ anymore. We will consider that $f$ captures the ambiguity of the task by providing several values as well. For instance, $f$ can be a diffusion model learning a distribution over $\mathcal{Y}$ for each input $x \in \Omega_{\mathcal{X}}$, and generating sample from it.

In the following, we will be only interested in the set of values that $f$ can take for each $x \in \Omega_{\mathcal{X}}$. When there is no ambiguity, we will denote $z \in \{f(x)\}$ to say that $z$ can be generated by $f$.

In this context the representation gap can be written

$$\mathcal{R}(\Omega, \Omega_f) = \int_{\Omega_{\mathcal{X}}} \int_{\Omega_x} \min_{z \in \{f(x)\}} \ell(y_x, z) \mathrm{d}x \mathrm{d}y_x .$$

Note that we recover the formula for non-ambiguous case when $\Omega_x$ is a singleton for each $x \in \Omega_{\mathcal{X}}$.

Using the insight from Proposition 3, we might want to assume that $f(x)$ takes values in the set $\mathbb{D}_x$ for $x \in \mathbb{D}_{\mathcal{X}}$, and that $\Omega_x$ is piece-wise constant outside of the training dataset.

Under this hypothesis, the model $f$ project the input $x$ toward the closest dataset input $x^* \in \mathbb{D}_{\mathcal{X}}$, and then generate a sample in the dataset target $\Omega_{x^*}$. More precisely, noting $x^* = \mathrm{argmin}_{x' \in \mathbb{D}} \ell(x, x')$, we have $\Omega_f = \{(x, y) | x \in \mathbb{D}_{\mathcal{X}}, y \in \Omega_{x^*}\}$, and $\{f(x)\} = \{(x^*, y) | y \in \mathbb{D}_x\}$.

However, we can see that such a model would have a very unstable behavior featuring many discontinuity as the density of the dataset input $\mathbb{D}_{\mathcal{X}}$ becomes high. Indeed, a typical case for real world datasets is that we have access to a single target $y_x$ for each covered input $x \in \mathbb{D}_{\mathcal{X}}$. Therefore the trained model $f$ would jump between modes for neighboring input $x \in \mathbb{D}_{\mathcal{X}}$ in the areas where $\Omega_x$ is multi-modal. This behavior is not what we observe in practice for trained neural network, so this hypothesis is not satisfying.

In order to escape this paradox, focusing again on diffusion models, we will consider that the $f$ is smooth both regarding the conditioning $x$ and the noise input $y_t$. It therefore project an input $(x, y_T)$ with initial noise $y_T$ toward a neighboring dataset point $(x, y) \in \mathbb{D}$.

We formalize this with the following hypothesis

### F.6 AMBIGUOUS CONDITIONAL TASKS AND COVERING NUMBER

The next Proposition extend the upper bound in 6 to the ambiguous task setting. We will restrict our attention to the Euclidean norm $\ell$.

**Proposition 8** (Conditional representation gap for ambiguous tasks). *Assume that $\Omega$ is a bounded Riemannian $d_\Omega$-manifold, that $\ell$ is the Euclidean norm, and that $f$ satisfies the smooth covering hypothesis 7. Then the representation gap of a dataset $\mathbb{D}$ of size $n$ by $f$ satisfies*

$$\mathcal{R}(\Omega, \Omega_f) \underset{n \to +\infty}{=} O\left(\frac{1}{n^{2/d_\Omega}}\right) . \tag{29}$$

*Proof.* This proof is two-step. First we prove that the representation gap can be reduce to the covering problem (i.e. finding an optimal covering of $\Omega'$ with ball of constant radius). Second, we derive an upper bound for this covering problem.

Let us first establish the link between the representation gap and the covering problem. Let $\varepsilon > 0$ and note $N(\varepsilon)$ the corresponding covering number of $\Omega$. For simplicity, we note $\mathcal{R}_n$ the minimum representation gap for a dataset $\mathbb{D}$ with $n$ points. Assume that $\mathbb{D}$ is an $\varepsilon$-covering of $\Omega$ with balls $B_1, \ldots, B_{N(\varepsilon)}$ centered on the dataset points $\mathbb{D}$.

Then, observe that under the smooth covering hypothesis 7, if $\varepsilon$ is small enough, for all $z \in \Omega$ we have $y^* \in \{f(x)\}$, where $z^* = \mathrm{argmin}_{z \in \mathbb{D}} \ell(z, z') = (x^*, y^*)$ is the point in the dataset $\mathbb{D}$ closest to $(x, y)$. We therefore obtain

$$\mathcal{R}(\Omega, \Omega_f) = \int_{\Omega_{\mathcal{X}}} \int_{\Omega_x} \min_{z \in \{f(x)\}} \ell(y_x, z) \mathrm{d}x \mathrm{d}y' = \int_\Omega \min_{z' \in \mathbb{D}} \ell(z, z') \mathrm{d}z .$$

Since $\mathbb{D}$ is an $\varepsilon$-coverage of $\Omega$, we have $\min_{z' \in \mathbb{D}} \ell(z, z')$ for all points $z \in \Omega$. Therefore, $\mathcal{R}_{N(\varepsilon)} \leq \mathcal{R}(\Omega, \Omega_f) \leq |\Omega|\varepsilon^2$.

Now, let us link the error $\varepsilon$ with the covering number $N(\varepsilon)$. It can be proved (see for instance Theorem 14.2 in Wu & Yang (2016)) that the covering number is bounded by

$$\left(\frac{1}{\varepsilon}\right)^d \frac{|\Omega|}{|B|} \leq N(\varepsilon) \leq \left(\frac{3}{\varepsilon}\right)^d \frac{|\Omega|}{|B|},$$

where we have noted $d = d_\Omega$, and $|B|$ the volume of the balls $B_k$. Noting $n \in \mathbb{N}$ and $\varepsilon = \frac{1}{n^{1/d}}$, we can rewrite this inequality as

$$\frac{|\Omega|}{|B|} n \leq N\left(\frac{1}{n^{1/d}}\right) \leq 3^d \frac{|\Omega|}{|B|} n.$$

Let $m = \left\lfloor 3^{-d} \frac{|\Omega|}{|B|} \right\rfloor$. Then we have $n \geq 3^d \frac{|\Omega|}{|B|} m \geq N\left(\frac{1}{m^{1/d}}\right)$. Since $\mathcal{R}_n$ is decreasing with n, we therefore deduce

$$R_n \leq R_{1/m^{1/d}} \leq |\Omega| \frac{1}{m^{2/d}} \leq \frac{|\Omega|}{\left(3^{-d} \frac{|\Omega|}{|B|} n\right)^{2/d}} = O\left(\frac{1}{n^{2/d}}\right)$$

This concludes the proof. $\qquad\square$

### F.7 MODEL INVARIANCE FOR AMBIGUOUS TASKS

We can attempt to generalize this result in the case of an equivariant model.

First note that in the case of a conditional ambiguous prediction task, both the input $x \in \mathbb{D}_\mathcal{X}$ and the output $y \in \mathbb{D}_\mathcal{Y}$ can be virtually augmented by a symmetry group. For instance, in the case of a conditional diffusion model $f$, the score function is typically conditioned by the input $x$. As the architecture for the conditioning model and the score function model may differ, each model may feature different equivariants. We denote $G_\mathcal{X}$ the symmetry group for the input $x \in \Omega_\mathcal{X}$ and $G_\mathcal{Y}$ the symmetry group for the target $y \in \Omega_\mathcal{Y}$.

We note $\pi_\mathcal{X}$ (resp. $\pi_\mathcal{Y}$) the quotient map from $\Omega_\mathcal{X}$ to $\Omega_\mathcal{X}/G_\mathcal{X}$ (resp. from $\Omega_\mathcal{Y}$ to $\Omega_\mathcal{Y}/G_\mathcal{Y}$). Then the invariance of $f$ simplifies the set $\Omega$ into $\Omega' = (\pi(x), \pi(y))|(x, y) \in \Omega$. Noting $d_{\Omega'}$ the dimension of $\Omega'$, we claim that the representation gap satisfies

$$\mathcal{R}(\Omega, \Omega_f) = O\left(\frac{1}{n^{2/d}}\right).$$

However, the rigorous proof of this statement and a more fine-grained analysis of the asymptotic behavior of the representation gap is left to future work.

### LLM USAGE

In this research, LLM have been used for polishing writing, discovery of related work (in particular for proof exploration), and code writing.

### REPRODUCIBILITY STATEMENT

Being mostly theoretical in nature, the results presented here are self-contained. Nevertheless, we provide source code to reproduce our Representation Gap implementation, along with an example demonstrating its use on MNIST, available on the Supplementary Material.

