# OpenReview forum: "Representation Gap: Explaining the Unreasonable Effectiveness of Neural Networks from a Geometric Perspective"
_ICLR.cc/2026/Conference — Submitted to ICLR 2026_

### Official Review · Reviewer_iokP · 2025-10-25

**Soundness:** 1
**Presentation:** 1
**Contribution:** 1
**Rating:** 2
**Confidence:** 2

**Summary:**

This paper propose to present an "explanation for the generalization properties" of neural networks, with theoretical contributions and an experimental validation.

They propose a metric called the "representation gap" which is the discrepency between the training data (e.g. manifold of (x,y) points) and the model predictions (e.g. (x, f(x)) predictions).

**Strengths:**

I'm failing to see the significance or novelty of this work, as described below.

**Weaknesses:**

Introduction:
--

The starting point of the discussion on generalization in NNs are (1) works on the implicit biases of gradient descent, and (2) "structural constraints [in diffusion models] to explain their impressive creativity".

However, (1) was shown to have very little influence on the generalization properties of NNs [1,2]

And (2) diffusion models are an odd choice to talk about structural constraints, as this point was made just as clearly much earlier, e.g. with CNNs.

[1] Loss Landscapes are All You Need: Neural Network Generalization Can Be Explained Without the Implicit Bias of Gradient Descent, Chiang et al. ICLR 2023

[2] Neural Redshift, Teney et al. CVPR 2024


The authors say:

> *the set of points (x, f(x)) that are reachable by a model f.*

This sounds a bit off: this is just the predictions of the model over the input space`?

--------------
Technical section
--

There are mentions of a relation with earlier work on manifold learning ("related work" section) and the manifold hypothesis (e.g. L770). I think there's a critical distinction between this work and the traditional manifold hypothesis that isn't stated. If I understand correctly, the latter is about the distribution of the data in the ambient input space $\mathcal{X}$, whereas this paper is focusing on the manifold defined by the data with its labels/predictions viewed as a manifold in $\mathcal{X} \times \mathcal{Y}$.

--------------

A central result is about equivariant architectures reducing the intrinsic dimensionality of the task. This seens trivial or perhaps tautological. The key assumption is that the enforced equivariance matches the symmetries present in the data. Therefore many possible points are equivalent under those symmetries, hence the intrinsic dimensionality of the data manifold is smaller, and the needs for training data (derived in the paper) are obviously improved.

--------------

The paper sets off trying to "explain generalization". The main results seem about the effect of enforcing hard invariances and equivariances in models. It's not clear how this helps with explaining generalization in a general sense, because most models don't have well-defined invariances, and the large body of work on enforcing such hard constraints in neural architectures has mostly been a failure (see e.g. Section 2 in [3]).

[3] Deep Learning is Not So Mysterious or Different, Wilson 2025

--------------

Name "representation gap": seems ill-chosen/misleading since this work is only considering the inputs and predictions of models, not its actual (internal/intermediate) representations.

--------------

Earlier work on dataset intrinsic dimensionality: not cited nor discussed. Here are some paper titles in order of relevance. The first in the list seems most important (e.g. related to the experiments with MNIST). The others may not be relevant enough to need to be cited, but I can't tell if this is why they're not cited, or if it's because the authors didn't know about this whole line of work.
- Intrinsic Dimensionality of Images and Its Impact on Learning
- Intrinsic Dimensionality of Image Representations
- Intrinsic dimension of data representations in deep neural networks
- Geometry of hidden representations of large transformer models
- Bridging Information-Theoretic and Geometric Compression in Language Models

--------------

L832: assumption of neural networks trained to interpolation with TV regularizer: does this match neural networks used in practice?

--------------
Empirical section
--

Empirical result in 5.1: "This is a remarkable result, since the models are trained with the generic L2 loss and have no knowledge of the structure of the data manifold."
On the contrary, the hard-coded equivariance is giving models full knowledge of this structure! This result is therefore fully expected. This experiments seems more like a "sanity check" that a demonstration of a "remarkable result".

--------------

L396 "Translation or rotation equivariance is added on top of the corresponding architecture*": how is this implemented?

--------------

Experiments "on real-world data" only use MNIST. Seems like a big claim for such a small experiment. It's fine not to have large-scale experiments and to focus on theoretical contributions, but the abstract/intro should not highlight "experiments on real-world data" as a contribution then.

--

There is a lot of math in the appendix that I didn't check given the much higher-level issues about the central claims of the paper mentioned above.

------------
Presentation
--

- The quality of the writing and presentation is subpar and makes the reading of the paper difficult.
- The bibliography is a complete mess. Most entries have no publication year, most have no venue, etc. Also please use correctly \citep and \citet.
- Missing sublot Fig.1(c) (mentioned in the caption)
- Figures 2 and 4 are identical???
- I don't understand the meaning of the first 4 words of the paper: "Implicit specification through data" (or what I'm guessing they mean makes no sense: this could apply to any learned model, it's not specific to NNs).
- L49 "*[quote fit without fear]*"??
- L79 ""**the** *standard technique to improve model performance*": what is **the** standard technique`?
- Appendix D "PREREQUISITE" -> "Preliminaries"
- L832 "minimal-norm hypothesis 6" -> "assumption 6"
- L478 "*could be leverage*" -> "*leveraged*"
- L485 "*distribution shift at test time, novelty introduction*": not sure what either of these mean; distribution shifts already refer typically to a shift between training and test time.
- L383 "*rotation-invariances*" -> no dash and "*rotational invariances*"

**Questions:**

I put a lot of questions above, but it is very unlikely that I will raise my score given its the many deficiencies at the time of its submission.

Overall this paper feels closer to a draft than a polished submission. I feel it borderline irrespectful to the community to submit half-finished papers for review.

In terms of contents, the main issues are that the main claims seem trivial (e.g. invariances) and/or well known (e.g. intrinsic dimensionality), and the empirical validation is straightforward yet claimed to show a "remarkable result", while being limited to simplistic toy data/MNIST. None of these issues would be a cause for rejection if these points were additions to another strong contribution. But here these points are all what the paper is about (unless I completely missed something major). For example, I don't have any problem with a weak experimental section if there was a strong conceptual contribution. Lastly, there is a disconnect between the technical contents of the paper and the stated goal at the beginning of the paper of "explaining generalization" (the paper basically shows that hard invariances/equivariances allow learning with less data, but this doesn't exist/usually doesn't work well in real architectures).

---

> ### Author Response · Authors · 2025-11-20
>
> We would like to thank the reviewer for their detailed comments, as well as to bringing our attention to several relevant related work. Moreover, we would like to apologize for the many typos of the submitted version. Unfortunately, an older draft was accidentally uploaded at the last minute, and we noticed the mistake only afterward. A corrected version will be provided.
>
> In the following, we address the main concerns raised in the review.
>
> > The starting point of the discussion on generalization in NNs are (1) works on the implicit biases of gradient descent, and (2) "structural constraints [in diffusion models] to explain their impressive creativity". However, (1) was shown to have very little influence on the generalization properties of NNs [1,2]
>
> The structural constraints we consider in our work arise from the architecture and the simplicity bias of neural networks (trained or untrained), a phenomenon well established theoretically and experimentally
> [1–6].
>
> Paper [1] argues that CNN generalization follows from the large volume (in the parameter space) of models having simple decision boundaries. But this not opposed to our claims. On the contrary, it corresponds to one of the key points of our work: generalization stems from simplicity bias, whether architectural [1,6] or induced by optimization [3-5]).
>
> The mentioned paper [2] suggests that MLPs and Transformers generalize due to architectural components (residual connections, layer normalization, specific activation functions), that bias them toward functions with a low complexity that often align with the target function. Again, this matches our claim that alignment between model and data symmetries explains generalization. We express this alignment through the representation gap -- the distance between the data manifold and the model manifold $\Omega_f$. In this view, the simplicity bias observed in [2] fits the assumptions of Theorem 3 (smooth model).
>
> > A central result is about equivariant architectures reducing the intrinsic dimensionality of the task. This seems trivial.
>
> The intuition relating equivariance to a reduction in manifold dimensionality is not new, but our work characterizes this dependency precisely. In particular, we obtain an **asymptotic equivalent** of the model performance as the number of samples grows. This contrasts sharply with most prior work, which provides only generalization or PAC bounds [7,8,9,10,11].
>
> We further argue that the representation gap offers the most natural and precise notion for formalizing this intuition.
>
> > The paper sets off trying to "explain generalization". The main results seem about the effect of enforcing hard invariances and equivariances in models.
>
> This paper is not focused only on equivariant neural networks. Our main focus is to introduce the representation gap, which is a novel concept that extends generalization error to the setting of generative modeling and ambiguous tasks, while being consistent in the traditional supervised prediction setting.
>
> Nevertheless, our main results highlight the dependency between asymptotic representation gap and network equivariance (Theorem 1 and 2) and more generally simplicity bias (Theorem 3), as we believe this is one of the main factor explaining neural network generalization.
>
> > And (2) diffusion models are an odd choice to talk about structural constraints, as this point was made just as clearly with CNNs.
>
> Diffusion models are particularly interesting because we can describe exactly the prediction space of a trained network (see Theorem 1).
>
> Moreover, our results also apply directly to CNNs, which were a prime motivation for this framework. By taking $G$ to be the group of translations, we can leverage the translation equivariance of CNNs and establish their sample efficiency, in terms of the intrinsic dimension. In fact, we can refine our results for CNNs by leveraging the locality of convolution (similarly to the treatment in [8]), further increasing the sample efficiency guarantees for this architecture.
>
> > Work non cited: [12,13]
>
> Paper [12] proposes an estimator of the intrinsic dimension of the data. However, its focus is quite different from ours. We provide a **theoretical** characterization of the neural network prediction space (Theorem 1) and of generalization (Theorems 2 and 3), including an **asymptotic equivalent** in the case of diffusion models (Theorem 2). In contrast, [12] only provides empirical results about generalization (see, e.g., Section 6.3).
>
> Our estimator of the intrinsic dimension (Section 5.3) appears as an indirect corollary of our theoretical results. It is also expected to be more robust in practice: the estimator of [12] can fail
> when sample size or density is low (see the bottom of p. 3), whereas ours converges rapidly,
> often with as few as $n=20$ samples. Moreover, the estimator in [12] requires a $k$ NN hyperparameter, introducing a bias-variance tradeoff, while ours requires only training data.

---

> ### Author Response · Authors · 2025-11-20
>
> In [13], although the authors also propose an estimator of intrinsic dimension, they do not relate it to neural network generalization, which is the central objective of the present work.
>
> > Work non cited: [14,15,16]
>
> The papers [14,15,16] estimate the intrinsic dimension of the latent space of neural networks, and is therefore quite different from our work. In particular, [15,16] are purely empirical.
>
> > L396 "Translation or rotation equivariance is added on top of the corresponding architecture": how is this implemented?
>
> It is implemented by input normalization and prediction shifting.
>
> [1] Loss Landscapes are All You Need: Neural Network Generalization Can Be Explained Without the Implicit Bias of Gradient Descent, Chiang et al. ICLR 2023
>
> [2] Neural Redshift, Teney et al. CVPR 2024.
>
> [3] Zhang, Chiyuan, et al. Understanding deep learning (still) requires rethinking generalization. Communications of the ACM 2021.
>
> [4] Li, Yue, and Yuting Wei. Minimum $\ell_ {1} $-norm interpolators: Precise asymptotics and multiple descent. arXiv 2021.
>
> [5] Ma, Chao, and Lexing Ying. On linear stability of sgd and input-smoothness of neural networks. NeurIPS 2021.
>
> [6] Kamb, Mason, and Surya Ganguli. An analytic theory of creativity in convolutional diffusion models. arXiv 2024.
>
> [7] Elesedy. Provably strict generalisation benefit for invariance in kernel methods. Neurips 2021.
>
> [8] Elesedy, Zaidi. Provably strict generalisation benefit for equivariant models, ICML 2021.
>
> [9] Chen, Dobriban, Lee. A group theoretic framework for data augmentation. JMLR 2020.
>
> [10]Tahmasebi, Jegelka. The exact sample complexity gain from invariances for kernel regression. Neurips 2023.
>
> [11] Petrache, M., & Trivedi, S. Approximation-generalization trade-offs under (approximate) group equivariance. Neurips 2023.
>
> [12] Pope, Phillip, et al. The intrinsic dimension of images and its impact on learning. arXiv 2021.
>
> [13] Gong, Sixue, Vishnu Naresh Boddeti, and Anil K. Jain. On the intrinsic dimensionality of image representations. IEEE/CVF 2019.
>
> [14] Ansuini, Alessio, et al. Intrinsic dimension of data representations in deep neural networks. NeurIPS 2019.
>
> [15] Valeriani, Lucrezia, et al. The geometry of hidden representations of large transformer models. NeurIPS 2023.
>
> [16] Cheng, Emily, Corentin Kervadec, and Marco Baroni. Bridging information-theoretic and geometric compression in language models. arXiv 2023.

---

> ### Comment · Reviewer_iokP · 2025-11-25
> **PDF**
>
> > we would like to apologize for the many typos of the submitted version. Unfortunately, an older draft was accidentally uploaded at the last minute, and we noticed the mistake only afterward. A corrected version will be provided.
>
> Pardon my incredulity, but I find this highly unlikely given the fact that the correct non-draft version still hasn't been uploaded. Uploading the correct PDF would have been my first reaction after realizing that the reviewers had seen an unfinished draft.

---

> ### Author Response · Authors · 2025-11-26
>
> We would like to thank again the reviewer for taking the time to read our work in detail and for providing us with many insightful comments. We have no doubt that they will help us improve the next version of the paper, especially concerning the presentation and the discussion of related work. We acknowledge the frustration of the reviewer and would like to apologize again for the typos and other mistakes that made the review process unnecessarily more difficult.
>
> Several concerns have been raised in the review, and we have tried to our best to answer the main points in our reply. We are grateful for the feedback that has already been given, but we are also looking forward for a deeper exchange during this discussion phase of the review process.
>
> We also inform the reviewer that an updated version of the paper has now been uploaded.

---

> > ### Comment · Reviewer_iokP · 2025-11-27
> > **Reviewer's response**
> >
> > Overall, the response does not address at all the concerns from my initial review.
> >
> > The questions and stated weakness in the initial review are not as much a request for more information in OpenReview messages, but an indication to the authors that there is important information missing from the paper.
> >
> > ---
> >
> > Just taking the first item of the response (not a critical flaw but as an example of the need for more than a message on OpenReview):
> >
> > >> The starting point of the discussion on generalization in NNs are (1) works on the implicit biases of gradient descent, and (2) "structural constraints [in diffusion models] to explain their impressive creativity". However, (1) was shown to have very little influence on the generalization properties of NNs [1,2]
> >
> > > The structural constraints we consider in our work arise from the architecture and the simplicity bias of neural networks (trained or untrained), a phenomenon well established theoretically and experimentally [1–6].
> >
> > The authors argue but contradict themselves with the contents of the paper, which is starting with "implicit biases of gradient descent", while the recent work on the simplicity bias they cite such as [1,2] shows precisely that gradient descent has little to do with it. Again this is a trivial issue in the introduction, but the same goes for all other issues stated in the initial review.
> >
> > ---
> >
> > As another example, the point of citing related work (cf. prior work about the intrinsic dimension of the data, reminding the authors that the intrinsic dimension is presented in the abstract as a central element of the proposed theory) is precisely to explain  to the reader how this contribution may be different.
> >
> > ---
> >
> > Another example:
> >
> > >> L396 "Translation or rotation equivariance is added on top of the corresponding architecture": how is this implemented?
> >
> > > It is implemented by input normalization and prediction shifting.
> >
> > Is this precisely described in the paper? I couldn't find it. I (and most other readers I expect) cannot make sense  of the experiments if the methods are not precisely described. Again, a few words on OpenReview do not replace clarity in the paper.
> >
> > ---
> >
> > Another example stateed in the initial review, and presented as an important corollary of the theory:
> > "*we prove that equivariant architectures improve performance by explicitly reducing this intrinsic dimension*".
> > I still don't see how this is not an obvious result (pretty much the definition of symmetries in the data). Either this statement is incorrect/incomplete, or I am missing something important, in which case I expect that other readers will too, meaning that there is some critical information that is missing unclear in the paper in its current form.
> >
> >
> > ---
> >
> > In conclusion, the overall issue is a mismatch between grand claims and the actual technical contributions of the paper (including: the goal of "*explaining generalization*", the new "*representation gap*" that does not involve representations, the "*remarkable empirical result*" that is remarkably obvious (see the initial review), etc.).
> >
> > I think the review provide numerous pointers that should help rework this submission into a contribution that is much more effective and informative for the readers. This will require substantial modifications, hence my rating/recommendation for rejection in its current form.

---

### Official Review · Reviewer_ZTQx · 2025-10-27

**Soundness:** 3
**Presentation:** 3
**Contribution:** 4
**Rating:** 8
**Confidence:** 2

**Summary:**

The paper studies the geometry of representations in deep neural networks and how that geometry connects to their generalization. The authors propose an eleganbt metric, the representation gap, and show how it is affected only by the intrinsic dimension of the task.

**Strengths:**

- The paper, despite its very technical nature, is generally very well written. The theoretical statements generally have intuitive/high-level explanations or informal versions
- The paper's subject is very relevant
- The experiments clearly back up the claims, and are generally illustrative
- I find the phrasing of the implication of Thm. 1 amazing and very insightful ("architecture = virtual augmentation")

**Weaknesses:**

I mostly have minor concerns.

- It is unclear to me whether (and how) this paper's contribution is related to the linear representation line of works by Victor Veitch et al. As far as I understand, they also take a geometric perspective (by defining a causal inner product). To better ascertain the novelty of the contribution, it'd be very useful to compare against to those works.
- *Abstract:* the two factors of generalization (and the related works) omit the large works on identifiable representation learning (particularly, nonlinear ICA), some of which results even formulate theoretical results for OOD (compositional) generalization.
- Eq (1): please explain in words what it means
- Thm. 3.: I think it'd be beneficial for the reader to better define what an "ambiguous task" means
- 5.1.: could you please add a bit more details how you could draw the two conclusions?
- 5.2.: please also elaborate the two observations as well
- Fig. 5: the label size is too small + please define what a quantizer means

### Minor points
- The citations do not show the year
- L049: a comment was presumable not removed after the citation
- L269: maybe it would be less confusing to denote constants $c$ and not $J$
- App. C.: since you list your assumptions, I'd rename this to "Assumptions"

**Questions:**

See above

---

> ### Author Response · Authors · 2025-11-20
>
> We would like to thank the reviewer for their detailed comments. In the following, we will try to address their main concerns.
>
> > How does this paper relate to the linear representation by Victor Veitch et al.
>
> The linear representation hypothesis [1] developed by Victor Veitch et al. posits that high-level concepts are represented as vector directions in some representation space and that linear algebra can be performed in this representation space.
>
> In [1] the geometry refers to linear manipulations in the latent space, while we refer to the geometry of the data distribution itself. Indeed, the aim in [1] is to perform probing and manipulation in the latent space of the LLM, while our focus is on explaining the creativity of neural networks. Moreover, the authors of [1] talk about representation as being described in the latent space, while we turn our attention to the prediction space of the model.
>
> > The related work omit the large works on identifiable representation learning (particularly, nonlinear ICA), some of which results even formulate theoretical results for OOD (compositional) generalization.
>
> Work on nonlinear ICA [2,3] focuses on the internal representation learned by neural networks. In this paper we argue instead that their creativity comes from their memorization capabilities and simplicity bias [4,5,6].
>
> The OOD compositional generalization results about non linear ICA (see for instance [7]) focus mostly on generalization bounds or PAC bounds (along with most literature on neural network generalization). Our work departs from this trend by providing precise **asymptotic equivalents**. This is in stark difference with prior work.
>
> > Eq (1): please explain in words what it means
>
> The representation gap in Eq. (1) measures how well the model’s prediction set $(X,f(X))$ approximates the data distribution $(X,Y)$.
>
> When $f$ outputs a single prediction, this reduces to the usual empirical risk: we compare $f(x)$ to $y$ and average over inputs.
>
> In the general case, however, $f$ may be generative or multi-prediction, so $f(x)$ is a set. We must therefore compare each target $y$ to the prediction set. Our choice is to project every sample $(x,y)$ to its closest point $(x',y')$ with $y'\in f(x')$ (this explains the infimum in Eq. (1)). This projection error is a special case of the Wasserstein distance (typically used to compare sets) and a natural generalization of quantization error.
>
> > Section 5.1.: could you please add a bit more details how you could draw the two conclusion?
>
> **First conclusion:** non-equivariant models converge
> toward the empirical distribution, so that $\Omega_f = D$. This can be seen in the upper row of Figure 3. Indeed, the samples generated by the model (pictured as blue dots) are localized exactly at the position of the dataset points (pictured as black crosses).
>
> **Second conclusion:** equivariant models converge towards the empirical distribution virtually augmented by the invariance group G. This can be seen in the lower row of Figure 3. Indeed, the samples generated by the model are localized at the position of the dataset points, but also at all translations (cube and wave datasets) and rotations (sphere dataset) of these points. This observation is formalized by Theorem 1, that states that $\Omega_f = G(D)$.
>
> > 5.2.: please also elaborate the two observations as well
>
> **First observation:** for all
> datasets, the three curves follow the same asymptotic evolution. This means that the representation gap predicted by the theory matches the empirical representation gap. It therefore confirms the accuracy of our theory.
>
> **Second observation:** we observe that the
> asymptotic evolution is independent of the dimension $d$ of the ambient space and depends only on the dimension $d\in\Omega$ of the manifold. For instance, this is observed on the cube dataset, where both dimensions can vary independently.
>
> > Fig. 5: define what a quantizer means
>
> The quantizers correspond to the dataset points. This is because the dataset points have been chosen to be optimal quantizers of the respective shape (cube, sphere or wave) in this experiment.
>
> > Thm. 3.: I think it'd be beneficial for the reader to better define what an "ambiguous task" means
>
> An ambiguous task refers to the case where several targets $y$ are equally acceptable for a given input x (see Section 4.2). For instance, for the task of image generation (with a diffusion model), several images can match a given input text prompt.

---

> > ### Author Response · Authors · 2025-11-20
> >
> > [1] Park, Kiho, Yo Joong Choe, and Victor Veitch. The linear representation hypothesis and the geometry of large language models. arXiv 2023.
> >
> > [2] Hyvarinen, Aapo, Hiroaki Sasaki, and Richard Turner. Nonlinear ICA using auxiliary variables and generalized contrastive learning. PMLR, 2019.
> >
> > [3] Khemakhem, Ilyes, et al. Variational autoencoders and nonlinear ica: A unifying framework. PMLR, 2020.
> >
> > [4] Kamb, Mason, and Surya Ganguli. An analytic theory of creativity in convolutional diffusion models. arXiv 2024.
> >
> > [5] Zhang, Chiyuan, et al. Understanding deep learning (still) requires rethinking generalization. Communications of the ACM 2021.
> >
> > [6] Shah, Harshay, et al. The pitfalls of simplicity bias in neural networks. NeurIPS 2020.
> >
> > [7] Buchholz, Simon, and Bernhard Schölkopf. Robustness of nonlinear representation learning. arXiv 2025.

---

> > ### Comment · Reviewer_ZTQx · 2025-11-24
> >
> > Thank you for your detailed response! I have a few follow-up questions.
> >
> > > The OOD compositional generalization results about non linear ICA (see for instance [7]) focus mostly on generalization bounds or PAC bounds (along with most literature on neural network generalization). Our work departs from this trend by providing precise asymptotic equivalents. This is in stark difference with prior work.
> >
> > 1. [7] does not study compositional generalization but model misspecification.
> > 2. **ICA does not formulate generalization bounds or PAC bounds, it proves asymptotic results**; thus, discussing the related literature should be part of the paper.
> > 3. For compositional generalization, see the works of, e.g., Sébastien Lachapelle, Jack Brady, and Thaddäus Wiedemer
> >
> > > The quantizers correspond to the dataset points. This is because the dataset points have been chosen to be optimal quantizers of the respective shape (cube, sphere or wave) in this experiment.
> >
> > In what sense is the quantizer optimal?

---

### Official Review · Reviewer_LR34 · 2025-10-29

**Soundness:** 3
**Presentation:** 3
**Contribution:** 2
**Rating:** 6
**Confidence:** 3

**Summary:**

This paper delves into the crucial problem of generalization in neural networks. The attempt to understand how design choices, like model architecture or training procedures, affect model behavior beyond the training data is genuinely intriguing. The authors propose a novel concept called the 'representation gap' ($\mathcal{R}_n$), moving away from traditional statistical learning theory frameworks like VC-dimension or Rademacher complexity. Instead, they offer a geometric perspective, measuring the discrepancy between the data manifold ($\Omega$) and the representation learned by the model ($\Omega_f$).

The core argument is that this representation gap asymptotically converges as $n^{-2/d}$, where $n$ is the dataset size and $d$ is the 'intrinsic dimension' of the task. This dimension $d$ is supposedly determined by the geometry of the data manifold and the symmetries (like equivariance) inherent in the model. I find the claim that techniques like equivariance provably improve generalization by reducing this intrinsic dimension particularly compelling. The analysis spans three scenarios: unconditional generative modeling (especially diffusion models), supervised prediction (assuming minimal-norm interpolation), and ambiguous prediction tasks.

Frankly speaking, the geometric approach itself is a refreshing and potentially insightful direction. Trying to explain the asymptotic nature of generalization performance with a single parameter—the intrinsic dimension—is quite ambitious. Providing a theoretical rationale for why equivariance works is also commendable.

However, I have some reservations. The theoretical development seems quite heavily reliant on specific assumptions about the models (mainly certain diffusion models or minimal-norm interpolators) and the data manifold. This makes me wonder about the broader applicability of these results to the messy reality of deep learning models and data. The concept of 'intrinsic dimension' $d$ is appealing, but the paper seems a bit lacking in discussing how one might estimate or know this value for complex, real-world data. While the experiments support the theory, they are mostly confined to low-dimensional synthetic data or MNIST, which might not be enough to demonstrate validity for more complex, high-dimensional modern problems.

**Strengths:**

1.  **Novel Perspective:** Offers a fresh geometric viewpoint on generalization, potentially moving beyond the limitations of classical statistical learning theory. The representation gap and intrinsic dimension concepts are thought-provoking.
2.  **Attempt at Unified Explanation:** The effort to connect implicit bias, structural invariants (like equivariance), and generalization through the concept of 'intrinsic dimension' is valuable.
3.  **Derivation of Asymptotic Equivalence:** Deriving the precise asymptotic convergence form ($n^{-2/d}$) for the representation gap, rather than just upper bounds, is a theoretically strong result.
4.  **Explanation of Equivariance:** Provides a clear and compelling theoretical explanation for the effectiveness of equivariant architectures – they reduce the task's intrinsic dimension $d$.

**Weaknesses:**

1.  **Strong Assumptions:** The theory seems to lean heavily on specific model classes (e.g., diffusion models satisfying certain conditions, minimal-norm interpolators) and geometric assumptions about the data lying on a Riemannian manifold. It's unclear how well these assumptions hold in typical deep learning scenarios. The claim that the learned representation $\Omega_f$ is either the dataset $\mathbb{D}$ itself ($\Omega_f = \mathbb{D}$) or its group-expanded version ($\Omega_f = G(\mathbb{D})$) might only apply under ideal convergence of diffusion models or strong invariance conditions.
2.  **Practicality of 'Intrinsic Dimension' $d$:** While theoretically interesting, the paper doesn't provide much guidance on how to practically estimate or determine the intrinsic dimension $d$ (either $d_\Omega$ or $d_{\Omega/G}$) for real high-dimensional data manifolds or quotient spaces. The post-hoc estimation of $d \approx 14$ for MNIST is shown, but its general applicability is questionable. This raises doubts about the practical utility of the theory's predictive power.
3.  **Limited Scope of Analyzed Models:** The focus is primarily on diffusion models and minimal-norm interpolators. How might this analysis extend to other important architectures, like standard CNNs or general Transformers?
4.  **Sensitivity of 'Representation Gap' $\mathcal{R}_n$ Definition:** The representation gap is defined using an integral of squared $\ell_2$ distances. Does the $n^{-2/d}$ scaling law hold if other loss functions or distance metrics are used? Could the result be overly specific to the $\ell_2^2$ metric?
5.  **Analysis of Ambiguous Prediction Tasks Seems Weaker:** The analysis for ambiguous prediction tasks (Section 4.2, Theorem 3) only provides an upper bound $O(n^{-2/d_\Omega})$ and relies on a somewhat vague "smooth covering model" assumption, making this part feel less solid compared to other sections.
6.  **Limited Experimental Validation:** The synthetic data experiments, while illustrative, are quite simple and low-dimensional. The MNIST experiment is interesting ($d \approx 14$), but more convincing real-world experiments on complex, modern deep learning problems are needed to truly validate this geometric perspective.

**Questions:**

1.  How valid do you believe the core assumptions of the theory (e.g., $\Omega_f = G(\mathbb{D})$ for diffusion models, minimal-norm interpolation, Riemannian manifold structure of data) are for complex, practical deep learning models (like ViTs, LLMs) and real-world data distributions? What are the limitations on the theory's scope if these assumptions don't hold perfectly?
2.  Are there practical methods to estimate or approximate the 'intrinsic dimension' $d$ ($d_\Omega$ or $d_{\Omega/G}$) for real-world tasks? If this dimension is unknown, how can the predictive power of the theory be utilized?
3.  Do you expect a similar $n^{-2/d}$ scaling law to hold if the representation gap were defined using a different distance metric (e.g., Wasserstein distance, KL divergence) instead of the squared $\ell_2$ distance? Or is this result specific to the $\ell_2^2$ metric?
4.  Could you elaborate on the "smooth covering model" assumption (Assumption 7) used in the analysis of ambiguous prediction tasks (Theorem 3)? Specifically, how do diffusion models satisfy this, and how might the results change if this assumption is violated?

---

> ### Author Response · Authors · 2025-11-20
>
> We would like to thank the reviewer for their detailed comments. In the following, we will try to address their main concerns.
>
> > What are the limitations on the theory's scope if its assumptions don't hold perfectly?
>
> The only hard constraint is that the model must be expressive enough to fit training data -- a condition that is typically satisfied in modern deep learning. As long as this holds, our theoretical results will apply. Our other assumptions are mostly technical, cover a wide range of experimental settings, and they are expected to hold in practice. They would only fail in very contrived theoretical scenarios, which are unlikely to arise in real applications.
>
> > Strong assumptions about the the models (minimal-norm interpolator) or the Riemannian data manifold, and the class of diffusion model. These may be misaligned with the messy reality of deep learning model and data.
>
> **Minimal-norm hypothesis.** We adopt this hypothesis because we believe that it most accurately captures the current practice in machine learning. Indeed, as discussed in the Introduction, recent theoretical work [1,2,3] show that large neural networks used in practice tend to converge toward minimal-norm interpolators of the training data.
>
> **Diffusion class.** For our results on diffusion models, we indeed focus on the DDIM class of diffusion models (note that this hypothesis concerns the diffusion forward and reverse equations, not the model architecture). Indeed, DDIM is widely used in practice, and most diffusion models can be seen as a variant of it. We expect that our theoretical analysis can easily be extended to other classes of diffusion models. Furthermore, our experiments indicate that the asymptotic behavior is the same for DDPM models, another widely used class of diffusion models.
>
> > Limited Scope of Analyzed Models: The focus is primarily on diffusion models and minimal-norm interpolators. How might this analysis extend to other important architectures, like standard CNNs or general Transformers?
>
> **CNNs.** Our assumptions cover the important case of CNNs, which are a specific instance of equivariant networks (translation equivariance). In fact, we can refine our results for CNNs by leveraging the locality of convolution (similarly to the treatment in [8]), further increasing the sample efficiency guarantees for this architecture.
>
> **Transformers.** The case of Transformer architectures is extremely important, but more challenging. It is the subject of our ongoing work. A promising starting point is to leverage the simplicity bias arising from many building blocks of this architecture (see for instance [11]).
>
> > Do you expect a similar scaling law to hold if the representation gap were defined using a different distance metric instead of the squared distance?
>
> Yes. The only difference would be the constants appearing in the asymptotic evolution (more precisely the volume $\lvert G\rvert$, $\lvert \Omega/G\rvert$ and the constant $J_d$ which depend on the metric).
>
> > How can we estimate the intrinsic dimension in realistic setting?
>
> Section 5.3 explains how the intrinsic dimension can be estimated through a simple linear regression on the representation gap curve. The example is shown on the MNIST dataset, but the same principle would apply to more general datasets. We observed this estimation method to be robust in our experiments.
>
> > Could you elaborate on the "smooth covering model" assumption (Assumption 7)? How might the results change if this assumption is violated?
>
> This assumption links the outputs $f(x)$ and $f(x')$ for two nearby points $x,x'$. It states that if $f$ has learned to generate a point $(x,y)$ in the training data, then it can generate the same target $y$ for a nearby input $x'$. Without this hypothesis (or a similar assumption about the model smoothness), the model $f$ could generate arbitrary outputs for input samples $x,x'$ which are arbitrarily close, making its generalization error impossible to control.
>
> This hypothesis is a natural consequence of the smoothness bias or neural networks (such as the minimal-norm bias or the simplicity bias), which has been shown theoretically and empirically across many practical settings [1,2,3,9,10]. For this reason, we expect it to hold for diffusion models. We expect our results to still hold if this assumption is violated, as long as some other continuity or smoothness assumption holds.

---

> > ### Author Response · Authors · 2025-11-20
> >
> > [1] Zhang, Chiyuan, et al. Understanding deep learning (still) requires rethinking generalization. Communications of the ACM 2021.
> >
> > [2] Li, Yue, and Yuting Wei. Minimum $\ell_ {1} $-norm interpolators: Precise asymptotics and multiple descent. arXiv 2021.
> >
> > [3] Ma, Chao, and Lexing Ying. On linear stability of sgd and input-smoothness of neural networks. NeurIPS 2021.
> >
> > [8] Kamb, Mason, and Surya Ganguli. An analytic theory of creativity in convolutional diffusion models. arXiv 2024.
> >
> > [9] Chiang, Ping-yeh, et al. Loss landscapes are all you need: Neural network generalization can be explained without the implicit bias of gradient descent. ICLR. 2022.
> >
> > [10] Shah, Harshay, et al. The pitfalls of simplicity bias in neural networks. NeurIPS 2020.
> >
> > [11] Neural Redshift, Teney et al. CVPR 2024.

---

> ### Comment · Reviewer_LR34 · 2025-11-27
>
> The authors have successfully answered most of my concerns. I will keep the score as is.

---

### Official Review · Reviewer_Lv4M · 2025-11-01

**Soundness:** 2
**Presentation:** 2
**Contribution:** 1
**Rating:** 2
**Confidence:** 4

**Summary:**

The paper investigates the approximation error for equivariant vs non equivariant neural networks, in the setting of diffusion models, showing that due to equivariant networks being constant on group action orbits, this lowers the intrinsic dimension, helping for better sampling bounds.

**Strengths:**

The main positive side is that the underlying principle shown is clearly presented and correct as far as I followed.

**Weaknesses:**

1) While the paper is aimed at equivariant neural networks, this is not stated explicitly enough, making it necessary to read it at least twice in order to follow the message.

2) I see a lot of the main principles of the paper are not new, and the paper just sets known results in the setting of diffusion models (which was not the focus of previous work). This may be due to authors being unaware of some important literature that contained very similar results (I say so because this literature is not cited). I will give only the main examples that come to mind.

a) For the role of "virtual augmentation" (theorem 1 of the paper under review), this principle was already shown in this work with similar ideas, with main difference being that the setup is not with diffusion models:
Elesedy. Provably strict generalisation benefit for invariance in kernel methods. Neurips 2021.
See also
Elesedy, Zaidi. Provably strict generalisation benefit for equivariant models, ICML 2021

b) For the dimension dependence (theorem 2), similar principles appear in
Chen, Dobriban, Lee. A group-theoretic framework for data augmentation. The Journal of Machine Learning Research, 21(1):9885–9955, 2020
See also
Tahmasebi, Jegelka. The exact sample complexity gain from invariances for kernel regression. Neurips 2023

c) For theorem 3, again results like this were present in the works in (b) and see also
Petrache, M., & Trivedi, S. (2023). Approximation-generalization trade-offs under (approximate) group equivariance. Neurips 2023

These are just a few works that contain similar results, and none of which are cited in this work, so I invite the authors to do a thorough literature search starting from these results, and to include a comparison with them.

**Questions:**

My main question is if the authors could present a detailed comparison to the results described in weakness 2, points a-b-c, highlighting if there are any new principles present, or if the main novelty compared to these is the application to diffusion models.

---

> ### Author Response · Authors · 2025-11-20
>
> We would like to thank the reviewer for their detailed comments, as well as for bringing our attention to several relevant related works. In the following, we will try to address their main concerns.
>
> > I see a lot of the main principles of the paper are not new, and the paper just sets known results in the setting of diffusion models
>
> **Conceptual novelty.**  This work introduces the representation gap, a novel concept that extends generalization error to the settings of generative modeling and ambiguous tasks, while remaining consistent with the traditional supervised prediction setting. It captures the geometric properties of both the data distribution and the prediction space, making it particularly suited to analyzing how model equivariance affects generalization.
> Even though the intuition relating equivariance and manifold dimension is not new, we believe our work provides the most natural concepts for formalizing and studying this relationship.
>
> **Different objects of study.** As a direct consequence, results from [1-5] concern different metrics (generalization error or generalization gap) while we study the representation gap. They are therefore quite distinct from our work and cannot be compared directly. Likewise, the notion of intrinsic dimension in [1] refers to the dimension of the function space, whereas we study the dimension of the data manifold. This is again a substantial conceptual difference.
>
> **Stronger guarantees.** The referenced literature provides only PAC bounds [3,5] or generalization bounds [1,2,4,5]. In contrast, we derive **precise asymptotic equivalents** of the models' generalization (see Theorem 2). Such results are significantly stronger and are rarely achieved in theoretical analyses of neural networks.
>
> **Wider task range and applicability.** The referenced works [2-5] focus exclusively on prediction tasks. Our results apply not only to prediction but also extend to generative models and multiple-prediction models (see Theorems 2 and 3). Furthermore, [1-4] focus on ridge regression while our results are far more general, covering prediction tasks, ambiguous prediction tasks and generative modeling (see Theorems 1, 2 and 3).
>
> **Wider model and architecture range.** The referenced papers [1,4] focus on kernel methods, and [2] studies linear models, while our results apply to arbitrary neural networks and are thus closer to the models used by practitioners in machine learning. This generality is enabled by fundamentally different proof techniques based on quantization theory (see Propositions 1,2 and 4 in Appendix). Moreover, [5] restricts attention to finite groups G, while our analysis naturally handles infinite groups commonly encountered in practice (such as rotation groups).
>
> **Compatibility with known results.** Among the referenced papers, [4] is the closest to our work, establishing a bound of the form $$\textrm{generalization error} ∝ $(\frac{\sigma^2vol(M/G)}{n})^(s/(s+d/2)).$$ This is consistent with the asymptotic behavior we derive in Theorems 2 and 3. However, [4] provides only an upper bound (while we obtain asymptotic equivalents in Theorem 2) and focuses on kernel ridge regression (while we cover the setting of generative modeling and ambiguous prediction as well).
>
> The similar asymptotic rate is expected: under mild regularity assumptions, it can be proved that the generalization error is bounded above and below by the representation gap. While both quantities share a decay of order $O(n^{-\alpha/d})$, comparing their leading constants is difficult, because they depend on different factors (e.g., the regularity of the function class $s$ in [4], versus the Zador constant $J_d$ in our case). A finer comparison would be an interesting direction for future work.
>
> > While the paper is aimed at equivariant neural networks, this is not stated explicitly enough.
>
> This paper is not limited to equivariant neural networks. It also considers a wider class of models, including general prediction models and smooth covering models (see Section C in Appendix). Nevertheless, our main results do highlight the dependency between asymptotic representation gap and model equivariance. We emphasize this aspect because we believe it captures one of the main factors underlying neural networks' generalization (see Theorems 2 and 3).
>
> [1] Elesedy. Provably strict generalisation benefit for invariance in kernel methods. Neurips 2021.
>
> [2] Elesedy, Zaidi. Provably strict generalisation benefit for equivariant models, ICML 2021
>
> [3] Chen, Dobriban, Lee. A group-theoretic framework for data augmentation. JMLR 2020.
>
> [4]Tahmasebi, Jegelka. The exact sample complexity gain from invariances for kernel regression. Neurips 2023.
>
> [5] Petrache, M., & Trivedi, S. (2023). Approximation-generalization trade-offs under (approximate) group equivariance. Neurips 2023.

---

> > ### Comment · Reviewer_Lv4M · 2025-11-27
> > **Partially convinced**
> >
> > Thank you for the explanations, I see now some good features of your proofs.
> > However I think that we did not converge on all points. Also, I mixed up [3] with another reference, corrected below (sorry).
> >
> > I'll use reference numbers as in your rebuttal.
> >
> > As I said, the main difference/novelty for me is that the paper includes generative / diffusion models.
> >
> > The Representation gap you introduce is closely related to generalization error, and to sampling bounds, **as described in your reply to referee oShr**, and for me as well, this comparison cannot but be in the back of my head while reading. I cannot help but see representation gap as a proxy towards generalization gap. Besides extending to generative models, can you explain how is representation gap a better notion than generalization gap?
> >
> > It is false that the work [5] only concerns finite groups, it concerns compact groups, including all compact Lie groups used in practice as well. **Do you claim that your work includes non-compact Lie groups?** I think this is not true, but correct me explicitly if this is wrong. You never mention any hypothesis on the group G or on the action (half joke: can you include groups of cardinality higher than the continuum? I believe not.. so please put some hypotheses somewhere)
> >
> > The bounds in [5] use doubling dimension, which is equal to usual manifold dimension in the case of Lie groups. The bounds in [5] use covering bounds (going back to Tikhomirov-Kolmogorov in this setting), a topic closely connected to quantization bounds (see the book by Graf-Luschgy that you cite in your paper, or even better, the proofs from [Gruber, P. M. (2004). Optimum quantization and its applications. Advances in Mathematics, 186(2), 456-497].)
> >
> > When I mentioned work [3] before, I was wrong, I meant another paper, sorry (I got confused because it had the same first author). I meant the sampling bounds from
> > [Chen, Z., Katsoulakis, M., Rey-Bellet, L., & Zhu, W. (2023, July). Sample complexity of probability divergences under group symmetry. In International Conference on Machine Learning (pp. 4713-4734). PMLR.]
> > I think that the bounds from the above work (instead of [3]) are also similar in spirit to your bounds. They are also good up to leading order, similarly to your bounds.
> >
> > I invite you to accept these similarities (to papers [1,2,4,5] mentioned earlier as well as the above by [Chen et al.]) for two reasons: (1) it is helpful for the reader (2) it is intellectually correct.
> >
> > The above works ideas do not seem strongly dependent on the model architecture either, even though the works may restrict themselves to some specific ones. I think that the bounds are similar in spirit to yours, and I find it correct to describe the similarity in detail, rather than avoid a sincere comparison.

---

### Official Review · Reviewer_oShr · 2025-11-07

**Soundness:** 2
**Presentation:** 2
**Contribution:** 2
**Rating:** 2
**Confidence:** 3

**Summary:**

This paper presents a novel geometric framework for analyzing generalization in neural networks, centered around a new metric termed the "representation gap." This metric quantifies the discrepancy (as measured by an integral of squared distances) between the true data manifold, $\Omega$, and the manifold effectively learned by the model, $\Omega_f$. The central theoretical contribution is an asymptotic scaling law for this gap: $\mathcal R_n \sim n^{-2/d}$, where $n$ is the dataset size and $d$ is the "intrinsic dimension of the task."

The authors provide a  partial theoretical explanation for the effectiveness of equivariant architectures, proving that they reduce this intrinsic dimension to that of the quotient space ($d = d_{\Omega/G}$), thereby accelerating the convergence of the representation gap in the asymptotic limit when $n \rightarrow \infty$.

The claims are supported by empirical validation on synthetic datasets and MNIST.

**Strengths:**

- The paper is overall well-written.
- The "representation gap" is an intuitive metric that generalizes empirical risk to the setting of manifold learning.
- The core idea of formalizing the sample efficiency gains of equivariance through a data-scaling law with a changing exponent ($d_{\Omega}$ vs. $d_{\Omega/G}$) is intriguing.

**Weaknesses:**

- In my view, the authors are overstating their results implications for generalization. For example, see l. 78: “We show that generalization is governed by the intrinsic dimension of the task, a single parameter which captures the difficulty level of the task, and may be directly linked to the data manifold geometry and the model invariants.”. Intuitively, the generalization gap might indeed be related to generalization, but this work provides no generalization bounds that rigorously connects those two. Obviously, if the representation gap goes to zero, the generalization error must also go to zero. It is however left completely unanswered what the decay rate of the generalization error is as a function of the rate of the representation error $\mathcal R_n$.

- The results w.r.t. representation gaps are fundamentally asymptotic, and it is not clear how close these asymptotic rates are to the ones in (realistic) finite sample regimes. In deep learning, it is well-known that typical models can memorize large training sets, and given enough samples, eventually perfectly learn any sufficiently regular distribution (due to uniform-convergence-type results). The actually important question is why these complex models manage to _efficiently learn_ natural distributions— a question which fundamentally requires finite-sample type results.

- The strength of distributional assumptions: see l. 61: “our analysis do not rely on any assumption about the data distribution $\rho$,
but only on the geometry of the manifold $\Omega$ on which this distribution is supported.” l. 92: “we do not make any assumption about the data distribution $\rho$ and adopt a geometric perspective instead.” In my view, knowing the support of a distribution is in fact a very strong assumption. I suspect that the authors intend to say that their results are governed by a single, informative quantity (the intrinsic dimension) that is very informative about the ground-truth distribution. However, saying that no distributional assumption is made seems a bit misleading.

**Questions:**

- See weaknesses.

- Do you think your results could be extended to the more realistic scenario of approximate symmetries in the data, and/or settings where the model's invariances do not exactly match the symmetries in the data?

Typos:
l. 49: "todo" marker

---

> ### Author Response · Authors · 2025-11-20
>
> We would like to thank the reviewer for their detailed comments. Below, we provide answers to the specific questions they raised.
>
> > This work provides no bounds that rigorously connects the representation gap and the generalization error.
>
> We thank the reviewer for bringing up this important point, which is indeed missing from our analysis. We address here the context of prediction tasks, where the generalization error admits a widely accepted definition, $$\mathcal{E}=\int_{\Omega}||y_x-f(x)||^2dx.$$
>
> In this setting, the representation gap simplifies to $$R(\Omega,\Omega_f)=\int_{\Omega} inf_{x'} ||(x,y_x)-(x',f(x'))||^2dx$$ and it is immediate from the infemum that $R(\Omega,\Omega_f)\leq\mathcal{E}$.
>
> Furthermore, if $f$ is $L$-Lipschitz, a simple trigonometry argument shows that $$\frac{1}{1+L^2}\mathcal{E}\leq R(\Omega,\Omega_f)\leq\mathcal{E}.$$
>
> Finally, if $R(\Omega,\Omega_f)=0$, then the integrand must vanish for all $x$ implying $f(x)=y_x$ almost everywhere, and hence $\mathcal{E}=0$.
>
> Overall, the eneralization error and representation gap are therefore closely related. We will incorporate this discussion in the paper.
>
> > The results w.r.t. representation gaps are fundamentally asymptotic, and it is not clear how close to the finite sample regime.
>
> Our theoretical results are indeed asymptotic. However, we have observed very fast convergence in practice: after $n=20$ samples for the sythetic experiments (see Figure 5) and $n=100$ sample for the MNIST experiment (see Figure 4).
>
> > Knowing the support of a distribution is a very strong assumption.
>
> Even though it is a strong assumption, it is considerably weaker than assuming full knowledge of the data distribution itself -- a basic assumption of the large majority of work on deep learning generaliation.
>
> > However, saying that no distributional assumption is made seems a bit misleading.
>
> We indeed do not make assumption about the data distribution itself, but only about the geometry of its support. We believe this is one of the main strengths of our approach, as assumptions about the full data distribution may fail to reflect the nature of real-world data -- for instance when the distribution evolves over time [1], samples are not i.i.d. [2], or sampling depends on the observer [3].
>
> > Could we extend the result to approximate symmetries and invariances?
>
> We thank the reviewer for raising this question, which is interesting but difficult to address. We expect that the analysis will depend sensitively on the details of the geometry of the manifolds $\Omega$ and $\Omega_f$. We also refer the reviewer to [4] for a partial answer concerning the generalization error in this setting.
>
>
> [1] Kuznetsov, Vitaly, and Mehryar Mohri. Generalization bounds for non-stationary mixing processes. Machine Learning 2017.
>
> [2] Mohri, Mehryar, and Afshin Rostamizadeh. "Rademacher complexity bounds for non-iid processes." NeurIPS 2008.
>
> [3] Burr Settles. Active learning literature survey. 2009.
>
> [4] Petrache, M., & Trivedi, S. Approximation-generalization trade-offs under (approximate) group equivariance. Neurips 2023.

---

### Meta-Review · Area_Chair_gWxx · 2025-12-26

**Summary:**

The paper builds its analysis around the “representation gap” and an associated intrinsic dimension, but these quantities are either unknown in practice or appear as hard to estimate as solving the learning problem itself. As a result, the theoretical statements — although mathematically appealing — offer limited operational guidance: it is unclear how one would use the framework to predict or improve generalization. Relatedly, many results rely on strong structural assumptions (e.g., precise manifold geometry, minimal-norm behavior, idealized diffusion dynamics), leaving their relevance uncertain. While the rebuttal clarified definitions and filled some technical gaps, as area chair I find that the core concerns — about the measurability of the introduced quantities and a concrete path forward using them — remain unresolved.

**Reviewer Concerns:**

Addressed:
The rebuttal clarified several definitions (representation gap, ambiguous tasks), added missing context on assumptions, and improved the explanation of how equivariance fits into the framework. It also acknowledged asymptotic nature of the results, added citations suggested by reviewers, and gave more intuition for some experimental choices.

Still outstanding:
However, key issues remain unresolved: the central quantities (representation gap, intrinsic dimension) remain effectively unmeasurable in realistic settings; the link between representation gap and generalization remains mostly heuristic; the theory still depends on strong structural assumptions whose scope is unclear; and it remains difficult to see what genuinely new understanding is gained beyond existing frameworks. These concerns limit both interpretability and the usefulness of the proposed perspective.

**Reviewer Scores:**

oShr (score = 2) – Rebuttal helped a bit, but core issues remained → likely unchanged.
Lv4M (score = 2) – Partially convinced but still noted unresolved similarities and gaps → maybe +1 at most, probably unchanged.
LR34 (score = 6) – Explicitly said they would keep the score → unchanged.
ZTQx (score = 8) – Engaged positively and stayed enthusiastic → unchanged.
iokP (score = 2) – Said clearly they would not raise the score → unchanged.

---

### Decision · Program_Chairs · 2026-01-26

Reject